



# Hygroscopicity and CCN potential of DMS derived aerosol particles

Bernadette Rosati[1,2], Sini Isokääntä[3], Sigurd Christiansen[1,4,5], Mads Mørk Jensen[1], Shamjad P. Moosakutty[1,6], Robin Wollesen de Jonge[1,7], Andreas Massling[8], Marianne Glasius[1], Jonas Elm[1], Annele Virtanen[3], and Merete Bilde[1]

[1]Department of Chemistry, Aarhus University, Denmark
[2]Faculty of Physics, University of Vienna, Austria
[3]Department of Applied Physics, University of Eastern Finland, Finland
[4]Faculty of Science and Technology, University of the Faroe Islands, Faroe Islands
[5]Department of Environmental Science, Stockholm University, Sweden
[6]Qatar Environment and Energy Research Institute, Hamad Bin Khalifa University, Qatar Foundation, Doha, Qatar
[7]Division of Nuclear Physics, Lund University, Sweden
[8]Department of Environmental Science, Aarhus University, Denmark

**Correspondence:** Bernadette Rosati (bernadette.rosati@chem.au.dk); Merete Bilde (bilde@chem.au.dk)

**Abstract.** Dimethyl sulphide (DMS) is emitted by phytoplankton species in the oceans and constitutes the largest source of naturally emitted sulphur to the atmosphere. The climate impact of secondary particles, formed through the oxidation of DMS by hydroxyl radicals, is still elusive. This study investigates the hygroscopicity and cloud condensation nuclei activity of such particles and discusses the results in relation to their chemical composition. We show that mean hygroscopicity parameters, $\kappa$,

during an experiment for particles of 80 nm in diameter range from 0.46 to 0.52 as measured at both sub- and supersaturated water vapour conditions. Ageing of the particles leads to an increase in $\kappa$ from for example 0.50 - 0.58 over the course of 3 hours (Exp. 7). Aerosol mass spectrometer measurements from this study indicate that this change most probably stems from a change in chemical composition leading to slightly higher fractions of ammonium sulphate compared to methanesulfonic acid (MSA) within the particles with ageing time. Lowering the temperature to 258 K increases $\kappa$ slightly, particularly for small

particles. These $\kappa$-values are well comparable to previously reported model values for MSA or mixtures between MSA and ammonium sulphate. Particle nucleation and growth rates suggest a clear temperature dependence, with slower rates at cold temperatures. Quantum chemical calculations show that gas-phase MSA clusters are predominantly not hydrated even at high humidity conditions indicating that their gas-phase chemistry should be independent of relative humidity.

## 1   Introduction

Aerosols originating from marine environments have a large impact on climate and are present over large parts of the Earth as oceans cover more than 70% of the Earths' surface (Seinfeld and Pandis, 2016). Marine aerosols affect climate not only by scattering and absorbing solar radiation, but also by acting as cloud condensation nuclei, CCN (e.g. IPCC, 2021). Marine aerosols consist of primary particles, mostly sea salt (Lewis and Schwartz, 2004) with various organic compounds (e.g. Facchini et al., 2008; Prather et al., 2013), and secondary particles comprising non-sea-salt sulphate originating from dimethyl sulphide

(DMS) oxidation. DMS is emitted to the atmosphere due to the decomposition of dimethyl sulfoniopropionate, which in turn,



is produced by phytoplankton in the oceans (Andreae, 1990). In addition, exposure of corals to air has also been found to result in emission of significant amounts of DMS into the atmosphere (Hopkins et al., 2016). DMS is considered as one of the most abundant natural sulphur compounds in the atmosphere (e.g. Seinfeld and Pandis, 2016), and the global flux to the atmosphere is estimated to be ∼28 TgS each year (Carpenter et al., 2012). In the atmosphere DMS reacts primarily with hydroxyl radicals

(OH) by either addition or hydrogen abstraction, where the relative importance of these two pathways has been shown to be sensitive to the prevalent temperature (Albu et al., 2006; Barnes et al., 2006). This reaction finally leads to compounds such as sulphuric acid ($H_2SO_4$, SA) and methanesulfonic acid ($CH_3SO_3H$, MSA) often found in the particle phase (Barnes et al., 2006; Mardyukov and Schreiner, 2018) as well as hydroperoxymethyl thioformate ($HOOCH_2SCHO$, HPMTF) (Veres et al., 2020; Ye et al., 2021).

Research on the importance of DMS for climate was stimulated by the so-called CLAW hypothesis (Charlson et al., 1987). The CLAW hypothesis proposes that higher ambient temperatures may increase the DMS production, which in turn, could enhance the formation of sulphate aerosol capable of acting as CCN. Subsequently, this could lead to changes in cloud micro-physical properties, coverage and lifetime, and therefore, cooling of the atmosphere. Although this hypothesis has been widely disputed, it has resulted in substantial progress in understanding the role of DMS oxidation products in the marine atmosphere

(e.g. Ayers and Cainey, 2007; Vallina and Simó, 2007; Smith, 2007; Quinn and Bates, 2011).

Hoffmann et al. (2016) investigated the multi-phase chemistry of DMS by advanced box-models and presented a new parametrization to include important aqueous-phase chemical processes. They showed that the atmospheric conversion of DMS into MSA is highly underestimated if the aqueous-phase chemistry of DMS is not considered. Their revised set of oxidation mechanisms was essential to explain MSA ratios found in atmospheric aerosols. Recently, the secondary aerosol formation

from DMS oxidation by hydroxyl radicals was studied in smog chamber experiments accompanied by kinetic multi-layer modelling (Rosati et al., 2021a; Wollesen de Jonge et al., 2021). It was observed that the secondary aerosol particle mass production from DMS oxidation was lower in experiments at humid conditions (relative humidity >50%) compared to dry ones (relative humidity <12%) for otherwise similar conditions of temperature, DMS and oxidant levels. In both cases the formed aerosol particles were found to consist mostly of MSA. Modelling and experimental data could be reconciled by including the newest

parametrizations for DMS oxidation extended by revised and complete multi-phase DMS oxidation mechanisms (Wollesen de Jonge et al., 2021).

Despite the significant contribution of DMS to atmospheric aerosols, knowledge about key properties especially concerning the hygroscopicity and CCN activity of DMS derived aerosols are sparse. Hygroscopicity, which describes the ability of a particle to take up water at subsaturated water vapour conditions, is one key aspect of the climatic effects of aerosols, and it

is predominately determined by the chemical composition of the particles. The CCN activity describes the potential of aerosol particles to be activated into cloud droplets at supersaturated water vapour conditions and it is known to depend on particle size and chemical composition (IPCC, 2021; Seinfeld and Pandis, 2016). The hygroscopicity (e.g. Berg et al., 1998; Nilsson et al., 2001; Lewis and Schwartz, 2004; Swietlicki et al., 2008; Modini et al., 2010; Fuentes et al., 2011) and ability to serve as CCN (e.g. Covert et al., 1998; Ovadnevaite et al., 2011; King et al., 2012; Prather et al., 2013; Collins et al., 2013, 2016;

Nguyen et al., 2017; Fossum et al., 2018; Rosati et al., 2021b) of marine aerosols and sea spray aerosol in particular has been



studied intensively. Few studies have addressed the water uptake behaviour of secondary DMS derived aerosols. Methanesulfonate (MSA) particles were scrutinised in a few studies: Johnson et al. (2004) reported experimentally derived hygroscopic growth factors (GF) for MSA at high humidity conditions, obtaining GF(90%) = 1.57 for particles with a dry diameter of 100 nm. Tang et al. (2019) investigated various particulate methanesulfonates and obtained hygroscopicity parameters in the range

of 0.37-0.47 based on CCN activity measurements (diameter: 40 - 60 nm) finding calcium methanesulfonate as the least and potassium and sodium methanesulfonate as the most hygroscopic compounds. The water cycle of sodium methanesulfonate, among others, have also been studied with an electrodynamic balance finding that particles deliquesce (i.e. dissolve completely) at RH=65.2-68.9 % and exhibit a GF(90%)=1.80 (Peng and Chan, 2001). Fossum et al. (2018) included the methanesulfonate anion to the list of ions considered in the Aerosol Inorganic-Organic Mixtures Functional groups Activity Coefficients (AIOM-

FAC) model (Zuend et al., 2008, 2011) to predict the CCN activity of methanesulfonate particles including MSA, sodium and ammonium methanesulfonate. Their calculations suggest that MSA and sodium methanesulfonate activate at similar critical supersaturations while higher supersaturations are needed to activate ammonium methanesulfonate particles of the same size.

To our knowledge the hygroscopicity and CCN activity of complex aerosols formed in the oxidation of DMS have not been addressed before. Therefore, the aim of this study is to investigate comprehensively the hygroscopic growth and CCN

activity of DMS-derived aerosols generated in an atmospheric simulation chamber and discuss these properties in relation to the chemical composition of the aerosols. Additionally, we show results of nucleation and growth rates of new particles formed through the oxidation of DMS with OH radicals. The experiments were designed to mimic a realistic marine atmosphere by simulating different relevant humidity and temperature conditions. For completeness, we also include in this work results from five experiments already presented by Rosati et al. (2021a) and/or Wollesen de Jonge et al. (2021) but not discussed with

respect to hygroscopicity and CCN activation potential previously. Our extensive measurement setup covers the formation rates of particles larger than 1.7 nm, the growth rates of 10-20 nm particles, the water uptake of particles from 10-150 nm (dry size), and provides insight into the bulk chemical composition of the DMS derived aerosols. Experiments were additionally complemented by quantum chemical modelling of MSA clusters.

## 2 Methodology

### 2.1 Theory

#### 2.1.1 Nucleation rate calculations

Particle nucleation rates were obtained using the method described in Tomicic et al. (2018). By measuring the concentration of particles with diameters exceeding a cut-off diameter of 1.7 nm ($N_{1.7}$), the nucleation rate $J_{1.7}$ is defined as:

$$J_{1.7} = \frac{dN}{dt} \qquad , \; N = \frac{N_{1.7}}{exp(-kt)} \qquad (1)$$

where $k$ denotes the rate of particle wall loss and $t$ the time after the initiation of new particle formation. The cut-off diameter of 1.7 nm was selected according to Kulmala et al. (2013) being described as the critical cluster size from where nucleation





occurs and thus $J_{1.7}$ constitutes an approximation of the nucleation rate. $J_{1.7}$ was obtained from the wall-loss corrected particle number concentrations, calculating the gradient between 20 % and 80 % of the maximum particle number concentration.

### 2.1.2 Water uptake by atmospheric aerosols

The water uptake behaviour of aerosol particles at sub- and supersaturated conditions can be described by the Köhler equation:

$$S = a_w \cdot exp\left(\frac{4M_w\sigma_s}{RT\rho_w D(RH)}\right), \tag{2}$$

where $S$ is the water vapour saturation ratio, $a_w$ the water activity, $\sigma_s$ the surface tension of the solution, $R$ the ideal gas constant, $T$ the absolute temperature, $\rho_w$ the density of water, $M_w$ the molecular mass of water and D(RH) the diameter of a
particle exposed to a certain relative humidity, RH.

At sub-saturated conditions of water vapour the hygroscopic growth factor at a given relative humidity (GF(RH) is calculated as the ratio of the particle diameter at high RH (D(RH)) to the dry particle diameter ($D_{dry}$):

$$GF(RH) = \frac{D(RH)}{D_{dry}}. \tag{3}$$

At supersaturated conditions of water vapour the critical supersaturation ($SS_{crit}$) needed for a specific particle size to be acti-
vated into a cloud droplet is in focus. This point is identified as the maximum of the Köhler curve (SS versus D(RH)).

To compare water uptake at sub- and supersaturated conditions, the hygroscopicity parameter $\kappa$, introduced by Petters and Kreidenweis (2007), can be used. By describing the water activity in terms of $\kappa$ in Eq. 2 and combining it with Eq. 3 the semi-empirical $\kappa$-Köhler equation can be obtained:

$$S = \frac{GF(RH)^3 - 1}{GF(RH)^3 - (1 - \kappa)} \cdot exp\left(\frac{4M_w\sigma_s}{RT\rho_w D(RH)}\right). \tag{4}$$

For a given supersaturation (or RH for the subsaturated case) a $\kappa_{GF}$ and $\kappa_{SS_{crit}}$ can be calculated.

### 2.2 Experimental Procedure

Experiments were carried out at the Aarhus University Research on Aerosol (AURA) atmospheric simulation chamber (Kristensen et al., 2017). In short, the facility consists of a 5 m$^3$ Teflon bag situated inside a temperature controlled room allowing experiments to be performed in the temperature range 257 to 299 K. The chamber is equipped with 24 UV lamps (wavelengths:
300-400 nm) located at the top and bottom of the bag. In high humidity experiments a custom-built humidifier was used to increase the relative humidity in the chamber to RH=70 %. $H_2O_2$ (30 % in $H_2O$, Merck, 1.07209.0250) was evaporated from a heated round-bottom flask (at 403 K) and flushed into the chamber using heated $N_2$ (333-343 K, 10 L/min). By photolysing $H_2O_2$, OH radicals were produced. The OH radical concentrations in the experiments are estimated to be in the range of typical tropospheric concentrations (4.64-5.71E+06 molecules cm$^{-3}$; Seinfeld and Pandis, 2016) based on supplementary experiments





investigating the decay of 1-butanol in the bag (Rosati et al., 2021a, b; Wollesen de Jonge et al., 2021). Either 200 or 400 ppb
of DMS (Sigma Aldrich, anhydrous ≥99.0 %, 274380) were injected into the chamber by evaporating a known amount and
guiding it into the chamber using 10 L min$^{-1}$ of N$_2$ gas heated to 333 K. The sequential order of the procedure was: 1) setting
the desired chamber temperature, 2) filling the chamber with purified air (Active Carbon, HEPA filter and zero-air generator
(Aadco Model 737-14)), 3) injecting humidified air to reach the desired RH, 4) injecting H$_2$O$_2$, 5) switching the UV lights on,
6) injecting DMS. The injection of DMS denotes the "time-0" in the experiments. Table 1 presents an overview of the experi-
mental conditions. Particle size distributions and chemical composition for Exp. 5 and 9 were further presented in Wollesen de
Jonge et al. (2021). This work focuses on the water uptake of the studied aerosol particles, thus Exp. 1, 5, 6, 7 and 9 are
presented in terms of the hygroscopicity and CCN activity, which were not addressed in any of the previous studies (Rosati
et al., 2021a; Wollesen de Jonge et al., 2021).

**Table 1.** Experimental conditions. $T_{ch}$ and $RH_{ch}$ denote the mean measured temperature and RH together with their standard deviations (1
sd) in the chamber. The DMS concentrations indicate the measured concentrations after injection except in Exp. 3, 6 and 7 marked with an
asterisk where theoretical values are given based on the liquid injected into the chamber as the PTR-ToF-MS was not available. Additionally,
particle growth rates (GR) obtained for the size range 10-20 nm are listed. See also Rosati et al. (2021a) for additional information regarding
Exp. 1, 6, and 7 and Wollesen de Jonge et al. (2021) for experiments 5 and 9. The last column specifies if any data of these experiments were
previously presented.

| Exp. number | Date | RH$_{ch}$ [%] | T$_{ch}$ [K] | DMS [ppb] | H$_2$O$_2$ [$\mu$L] | GR [nm h$^{-1}$] | |
|---|---|---|---|---|---|---|---|
| 1 | 13.02.2019 | 5.2 ± 0.2 | 292.6 ± 0.1 | 120 | 418 | 13.1 ± 0.6 | see also Rosati et al. (2021a) |
| 2 | 15.02.2019 | 63.0 ± 5.5 | 292.4 ± 0.4 | 150 | 418 | 24.8 ± 3.8 | |
| 3 | 05.03.2019 | 62.2 ± 1.2 | 292.8 ± 0.3 | 200* | 1500 | 20.2 ± 0.5 | |
| 4 | 22.02.2019 | 58.5 ± 1.8 | 292.4 ± 0.4 | 310 | 418 | 20.5 ± 0.9 | |
| 5 | 26.02.2019 | 56.4 ± 4.4 | 292.4 ± 0.4 | 330 | 1500 | 26.2 ± 1.9 | see also Wollesen de Jonge et al. (2021) |
| 6 | 04.03.2019 | 13.7 ± 1.7 | 292.8 ± 0.1 | 400* | 1500 | 15.9 ± 1.1 | see also Rosati et al. (2021a) |
| 7 | 07.03.2019 | 9.2 ± 0.6 | 292.8 ± 0.2 | 400* | 1500 | 15.5 ± 0.5 | see also Rosati et al. (2021a) |
| 8 | 18.02.2019 | 73.2 ± 5.7 | 273.2 ± 0.8 | 170 | 418 | 16.3 ± 0.7 | |
| 9 | 01.03.2019 | 75.5 ± 4.1 | 273.1 ± 0.8 | 220 | 1500 | 11.9 ± 0.4 | see also Wollesen de Jonge et al. (2021) |
| 10 | 20.02.2019 | 70.7 ± 5.1 | 259.3 ± 1.3 | 120 | 418 | 4.5 ± 0.0 | |
| 11 | 28.02.2019 | 75.3 ± 1.0 | 258.8 ± 0.2 | 280 | 1500 | 7.7 ± 0.9 | |

## 2.3 Instrumentation and Analysis

### 2.3.1 Stationary instrumentation

Nitrogen oxides (NO$_x$) and ozone (O$_3$) levels were continuously measured with a chemiluminescent monitor (AC32M, Envi-
ronnement S.A) and a UV absorption ozone analyser (O342 Module, Environment S.A), respectively. The temperature (T) and
RH were monitored in the centre of the bag (HC02-04 sensor; Rotronic AG, Switzerland).





### 2.3.2 Volatile organic compounds

During Exp. 1, 2, 4, 5 and 8-11 a Proton Transfer Reaction Time-of-Flight Mass Spectrometer (PTR-ToF-MS 4000, Ionicon Analytik, Innsbruck, Austria; e.g. Graus et al., 2010; Cappellin et al., 2012; Sekimoto et al., 2017) was employed to continuously monitor DMS concentrations. The instrument is based on a soft-ionisation technique using hydronium ions ($H_3O^+$) to ionise gas-phase compounds with proton affinities equal to or greater than that of water, which is the case for most volatile

organic compounds (VOC). After the reaction with the primary ion occurs in the drift tube, the ionised compounds are detected in the ToF-MS. The PTR-ToF-MS was run with drift tube conditions of 650 V, 80°C and 2.4 mbar and thus an E/N number of 143 Townsend. The instrument was connected to the chamber via a 2 m long Teflon tube, wrapped in heating tape set to 60°C, that was connected to the heated inlet of the instrument (temperature set to 80°C). Additionally, the instrument flow rate was increased by adding a sheath flow of 200 mL min$^{-1}$ to reduce sampling losses.

Data analysis was performed using the PTR-MS Viewer 3 (Ionicon). Prior to the measurement campaign a transmission calibration was carried out based on specific reaction rate constants and mass discrimination factors as described elsewhere (Taipale et al., 2008; Liu et al., 2018).

### 2.3.3 Particle number concentration

An Airmodus A11 nano Condensation Nucleus Counter (nCNC, Airmodus), consisting of a Particle Size Magnifier (PSM,
Airmodus A10) and a Condensation Particle Counter (CPC, Airmodus A20), was used to measure particles above 1.7 nm in diameter during Exp. 5, 9 and 11. A thorough description of the instrument is given elsewhere (Vanhanen et al., 2011). In short, the PSM was operated with diethylene glycol as working fluid. Supersaturated diethylene glycol leads to growth of the smallest (approximately 1 nm) particles by condensing the working fluid onto them. When the particles reach sizes of approximately 90 nm they are guided to a condensation particle counter (using butanol as working fluid) and optically detected therein. The
PSM was operated in fixed mode, counting all particles above the selected 1.7 nm threshold. The saturator flow rate was kept at 0.28 L min$^{-1}$ in order to obtain the desired cut-off diameter.

### 2.3.4 Particle size distributions

A Scanning Mobility Particle Sizer (SMPS, TSI model 3938) was used to measure the dry particle number size distributions. The Electrostatic Classifier (EC, TSI model 3082) was either connected to a nano Differential Mobility Analyser (DMA; TSI
model 3085A) for sizes between 2 and 65 nm or a long DMA (TSI model 3081) for the size range 10-422 nm. Initially the nano DMA was used and switched to the long DMA when particles grew out of the size range. The DMAs were connected to a nano Water-based Condensation Particle Counter (WCPC, TSI model 3788). The SMPS data analysis was performed with the Aerosol Instrument Manager (AIM) version 10.2.0.11 using the corrections for diffusion losses and multiple charges. A sheath to aerosol flow ratio of 10:1 was chosen (Nano SMPS: 15:1.5 L min$^{-1}$; Long SMPS: 6:0.6 L min$^{-1}$). A silica gel diffusion
dryer was used to ensure a RH<10% at the inlet of the SMPS.





In order to retrieve particle properties, such as hygroscopicity, representative for the main particle mode, we used the size distributions measured with SMPS over time. A log-normal distribution was fitted to the SMPS data at every time step. The mean and standard deviation of the particle number size distribution were retrieved from the fit. A particle size was classified as representative for the major particle population in the chamber when it was within the fitted mean $\pm$ 1 sd of the SMPS size distribution. All mean values presented in the manuscript refer to the time when particle properties were measured within the main growth-mode. An example for the limits found for Exp. 5 is shown in Fig. S1 in the supplementary information (SI).

### 2.3.5 Water uptake at sub- and supersaturated conditions

The hygroscopic growth of 10-150 nm (dry size) particles was measured with two separate Humidified Tandem Differential Mobility Analysers (HTDMA) at a constant RH of 80 %. In brief, an HTDMA consists of two DMAs, a CPC and a humidifier in-between the two DMAs. The first DMA is used to select a monodisperse sample from the dry polydisperse particle population, which is then exposed to a selected RH. The size distribution of the humidified particles is then measured with the second DMA followed by the CPC. In this way the hygroscopic growth factor GF(RH) can be retrieved (Eq. 3).

The hygroscopic growth of particles with $D_{dry}$=10, 15 and 20 nm particles was measured with a custom built "nano-HTDMA" described in more detail e.g. in Keskinen et al. (2011) and Tikkanen et al. (2018). In short, the nano-HTDMA system consists of two nano-DMAs (TSI, model 3085) and a water based CPC (TSI, model 3785) to count the particles. A silica gel diffusion drier was placed in front of the first DMA to make sure the particles entered the instrument with RH<10 %. The DMAs were operated in an open-loop system, with a sample flow of 1 L min$^{-1}$ and a sheath flow of 10 L min$^{-1}$. To ensure stable conditions through the particle sizing, the sample flow and the sheath air inside the second DMA were humidified separately. The measured GFs were then corrected to the target RH of 80 % assuming a constant hygroscopicity parameter $\kappa$ within small deviation around the target humidity ($\pm$1.5 %), as in Tikkanen et al. (2018). Ammonium sulphate particles (Sigma Aldrich, purity $\geq$99.0 %) were used to calibrate the instrument for offsets in dry size selection. After taking into account the uncertainties of all devices influencing particle sizing, the uncertainty in the GFs can be estimated to be at maximum 2.5 %.

The hygroscopic growth of particles with $D_{dry}$=30, 50, 80, 100 and 150 nm was measured with a commercially available HTDMA (Brechtel, Model 3002, Lopez-Yglesias et al., 2014), from now on termed "long-HTDMA". A diffusion drier at the inlet ensured that particles entered the instrument at RH<10 %. The aerosol flow of the first and second DMA were 0.8 and 0.5 L min$^{-1}$, respectively, while the sheath flows were 5 L min$^{-1}$ in both systems. The instrument was calibrated using ammonium sulphate (Sigma Aldrich, purity >99.95 %) particles. The overall uncertainty in GF(80 %) is estimated to be below 3 %, based on the combination of instrumental uncertainty (estimated to be 2 % sizing accuracy and 1 % RH uncertainty) and comparison to theoretical values calculated with Köhler theory (max. 2 % for the chosen size range).

The ability of particles to act as cloud condensation nuclei (CCN) was monitored with a Continuous-Flow Streamwise Thermal-Gradient CCN Chamber (CFSTGC, CCN-100 from Droplet Measurement Technologies; Roberts and Nenes, 2005). The CCN instrument was operated following the Scanning Mobility CCN Analysis (SMCA) described in Moore et al. (2010) to increase the temporal resolution. In short, we measured the critical dry diameter, $D_{p,c}$ of mobility size-selected aerosols. Aerosol laden air from AURA was dried using diffusion dryers (RH<10 %), before entering the Electrostatic Classifier. The



classifier consisted of a long-DMA (TSI model 3081) and an aerosol neutraliser (X-ray source, TSI model 3087). A scanning voltage was applied (scan time 120 s up and 15 s down) and the mono-disperse outlet stream of the DMA was directed to both the CCNc and CPC (TSI model 3010). Time was synchronised daily. The CCN supersaturation was initially set high ($\sim 1.4\%$), since the newly formed aerosols were small in the beginning. As the particle size distribution grew to larger sizes, the supersaturation of the CCNc was decreased to $\sim 0.2$ %, which is representative for supersaturations found in the marine environment (Ditas et al., 2012). Scan time alignment, multiple charge correction, quality control, sigmoidal fitting of the activation curves (CCN/CN vs. $D_p$) was done using the "SMCAProcessor.xls" file available at http://nenes.eas.gatech.edu/ Experiments/SMCA.html (Accessed February 2019). The supersaturation of the CCN column was calibrated on the 11.03.2019 using atomised ammonium sulphate (99.9999 % purity, 33 mg in 500 mL MilliQ). The measured $D_{p,c}$ was used as input to the "Köhler curves" module of the Extended AIM Aerosol Thermodynamics Model to estimate the supersaturation (http: //www.aim.env.uea.ac.uk/aim/kohler/kohler2.php, accessed March 2019). A schematic of the SMCA calibration setup and a calibration curve performed with ammonium sulphate are shown in Fig. S2 and Fig. S3 in the SI.

### 2.3.6 Chemical composition of particles

A High-Resolution Time-of-Flight Aerosol Mass Spectrometer (HR-ToF-AMS, Aerodyne Research Inc.) was used to measure the chemical composition of the particles in real time. The instrument setup is described in detail in Decarlo et al. (2006). The instrument was employed as described in Rosati et al. (2021a). HR-ToF-AMS data were processed in Igor Pro 8 by the data analysis software packages SQUIRREL (version 1.62) and PIKA (version 1.22). The PIKA default HR-ToF-AMS collection efficiency (CE) of 1 was used. The MSA calibration as presented in Rosati et al. (2021a) was applied to quantify particulate MSA. Unit-mass resolution efficient Particle Time-of-Flight (ePToF) data was exported for the MSA tracer ion *m/z* 79 and sulphate tracer ion *m/z* 80. This sulphate tracer ion was selected because MSA only contributes $0.04 \cdot [m/z\ 79]$ to the *m/z* 80 signal. The relationship between vacuum aerodynamic diameters as retrieved from ePToF data and mobility diameters for spherical particles, used in the measurement of the water uptake, can be calculated according to DeCarlo et al. (2004):

$$\rho_p = \frac{d_{va}}{d_m}\rho_0, \tag{5}$$

where $d_{va}$ is the vacuum aerodynamic diameter, $d_m$ is the electric mobility diameter, $\rho_0$ is standard density (1 g cm$^{-3}$) and $\rho_p$ is the density of the particle. The particles were assumed to have no voids, which results in the density of the particle to be equal to the density of the material. PToF time series were smoothed using a 15-point (i.e. 15 minutes) moving average to decrease short-term fluctuations.





## 2.4 Quantum Chemical Calculations

### 2.4.1 Computational Details

The Gaussian09 (Frisch et al., 2013) program was used to obtain the molecular cluster structures and calculate the vibrational
frequencies. The $\omega$B97X-D (Chai and Head-Gordon, 2008) functional was employed as it has been shown to exhibit low errors
in the binding energies of atmospheric molecular clusters compared to higher level CCSD(T) calculations (Leverentz et al.,
2013; Elm et al., 2013; Elm and Kristensen, 2017). The calculations were performed using the 6-31++G(d,p) basis set, which
has shown good agreement with basis sets of significantly larger sizes (Elm and Mikkelsen, 2014; Myllys et al., 2016).

The ORCA program version 4.0.0 (Neese, 2012) was used to calculate the single point energy (i.e. the electronic energy of
a molecule for a certain arrangement of the atoms in the molecule) of all the clusters using DLPNO-CCSD(T$_0$) (Riplinger and
Neese, 2013; Riplinger et al., 2013) with an aug-cc-pVTZ basis set. The aug-cc-pVTZ/C and aug-cc-pVTZ/JK auxiliary basis
set were used for density fitting and Coulomb/exchange fitting. We have recently demonstrated that this level of theory yields
total and relative binding energy results in good agreement with higher level explicitly correlated coupled cluster calculations
on a test set of 45 cluster structures (Schmitz and Elm, 2020) and sulphuric acid hydrates (Kildgaard et al., 2018a). All the
thermochemical parameters have been calculated at 298.15 K and 1 atm.

### 2.4.2 Cluster Structure Sampling

We have recently reported the application of a systematic sampling technique to obtain the molecular structures of sulphuric
acid - water clusters (Kildgaard et al., 2018a) and carboxylic acid - water clusters (Kildgaard et al., 2018b). Here we utilise the
same method to thoroughly sample the rich potential energy surface of the methanesulfonic acid (MSA) - water clusters. The
following procedure was applied:

1. Water molecules are placed around all the exterior atoms of the cluster.

2. Generated conformers are optimised using the PM6 (Stewart, 2007) method.

3. Identical conformers were excluded based on energy and rotational temperature.

4. The Gibbs free energy is calculated at the $\omega$B97X-D/6-31++G(d,p) level of theory.

5. All conformers below 3 kcal/mol are then used as input structures for the addition of the next water molecules (step 1).

This approach generates a massive amount of conformers (350, 1084, 2027, 6801 and 5542 for the respective hydrates) and
will yield a qualified guess for the global minimum cluster structure.





## 3 Results and Discussion

### 3.1 Nucleation and Growth Rates at Different Temperatures

Figure 1a presents particle number concentrations and nucleation rates of particles larger than 1.7 nm in diameter at 258 K (in blue), 273 K (in orange) and 293 K (in red) measured during Exp. 5, 9 and 11, respectively. The highest particle number concentrations were recorded at 298 K, and the lowest ones at 258 K. Nucleation rates increased with increasing temperature, e.g. from 68.7 cm$^{-3}$s$^{-1}$ at 258 K to 426.3 cm$^{-3}$s$^{-1}$ at 293 K. New particle formation was initiated at an earlier point in time at 293 K (immediately after DMS injection) compared to 258 K (approximately 10 minutes after DMS injection).

Figure 1b shows particle growth rates (GR) of particles with diameters between 10 and 20 nm, calculated using the maximum-concentration method (Lehtinen and Kulmala, 2003; Kulmala et al., 2012). The overall GR range (4.5 - 26.2 nm h$^{-1}$; see values in Table 1) fits well with ambient values measured in marine and coastal areas (1.8 - 20 nm h$^{-1}$; Manninen et al., 2010; Yli-Juuti et al., 2011). Similar to the nucleation rates, GR increase with increasing temperature. The colour code in Fig. 1b denotes the RH during the experiments. Results at 293 K show that lower RH values lead to lower GR compared to
experiments performed at high RH.

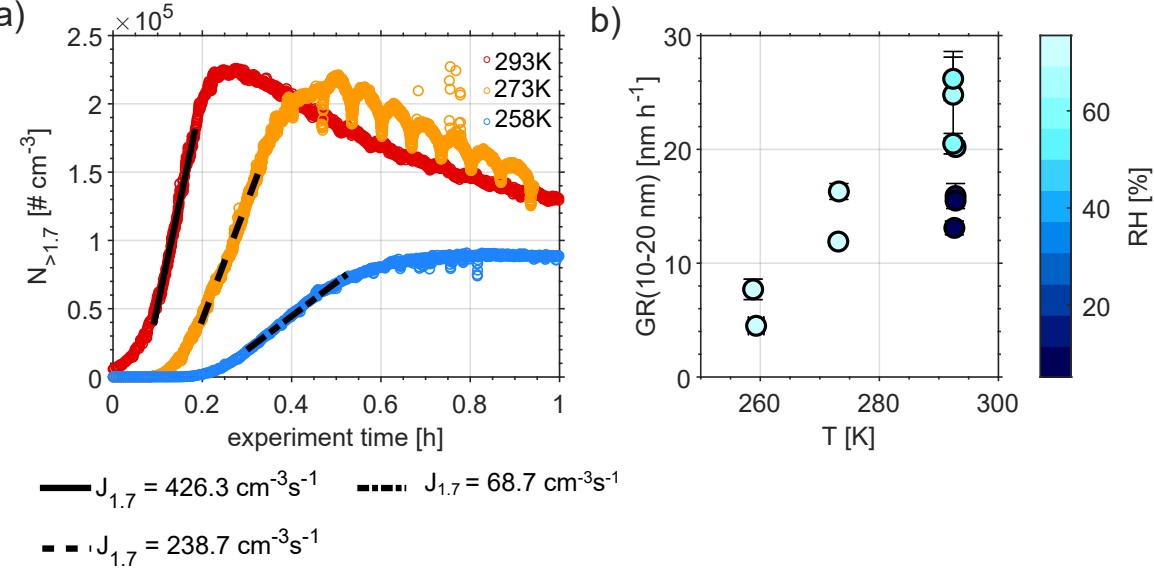

**Figure 1.** a) Nucleation rates (J) of particles larger than 1.7 nm during Exp. 5, 9 and 11 and b) growth rates (GR) for the size range from 10 to 20 nm at three different temperatures during Exp. 1-11. The fluctuations seen at T=273 K are due to the fact that the PSM was set to measure in scanning mode after a while.

    The decrease of nucleation and growth rates with decreasing temperature is consistent with a lower observed loss rate of DMS at lower temperatures, see Fig. S4 and Table S1 in SI illustrating an example for each of the temperatures investigated in this study and a comparison between dry and humid experiments (Exp. 1, 2, 8 and 10). This trend is in line with the





predicted temperature dependence obtained with the Arrhenius expression used in the Master Chemical Mechanism version

3.3.1 (MCMv3.3.1) and Hoffmann et al. (2016) delineating slower reaction rates at colder temperatures. The decay of the concentration of DMS over time for similar experiments at 293 and 273 K, dry and humid conditions, was formerly presented in Rosati et al. (2021b) and Wollesen de Jonge et al. (2021). Wollesen de Jonge et al. (2021) found good agreement between measured and modelled decay rates using the ADCHAM box model. Interestingly, opposite temperature trends were found by e.g. Albu et al. (2006) and Barnes et al. (2006).

Nucleation and growth rates are known to be inherently dependent on temperature and RH (Seinfeld and Pandis, 2016). Typically at lower temperatures both nucleation and growth rates increase as the Gibbs free energy barrier decreases. This has also been observed in laboratory experiments with homogeneous nucleation of sulphuric acid and water at temperatures between 207 - 299 K and RH between 11 and 58 % (Duplissy et al., 2016). Ambient measurements of nucleation and growth rates, however, show ambiguous trends probably related to different temperature dependent processes occurring at the same

time (Kerminen et al., 2018). A possible explanation for the decreasing nucleation and growth rates with decreasing temperatures seen in this study can be that the oxidation of DMS is slower at colder temperatures, thus fewer volatile compounds are available for cluster formation as they are formed more slowly. This is also supported by the above presented DMS decay trends illustrating slower decay rates with colder temperatures. The absolute water concentration could also have an effect on the observed trends in the nucleation rate. Water has been shown to increase the nucleation rate if the initial cluster is

weakly bound such as for sulphuric acid – ammonia clusters (Henschel et al., 2014b, 2016). As the initial cluster formation in our chamber setup most likely consist of relatively weakly bound sulphuric acid – methane sulfonic acid – ammonia clusters (Rosati et al., 2021a), the higher absolute concentration of water at colder temperatures could also influence the observed trend in the nucleation rates.

   OH concentrations in the chamber were obtained from the decay of 1-butanol due to reactions with OH radicals produced by

photolysing $H_2O_2$. 1-butanol decay rates and resulting OH concentrations for experiments performed at the same temperature and humidity conditions as used in the herein presented DMS experiments are illustrated in Fig. S5 and Table S2 in the SI. The herein found OH concentrations are in the range of tropospheric levels (e.g. 1-2E+06 molecules cm$^{-3}$ Seinfeld and Pandis, 2016; 1.09E+06 molecules cm$^{-3}$ Li et al., 2018).

## 3.2 Hygroscopicity, CCN activity and Chemical Composition

### 3.2.1 Effect of Atmospheric Ageing at 293 K

Atmospheric ageing of DMS derived secondary aerosols was simulated by prolonged exposure to OH radicals at different temperatures and RH. We first consider the water uptake at sub- and supersaturated water vapour conditions at 293 K for both dry and humid conditions. Figure 2 illustrates the water uptake behaviour in the form of $\kappa$-values obtained from hygroscopic growth and CCN measurements overlaid on particle number size distributions during Exp. 5 (humid conditions) and Exp. 7 (dry

conditions). In both cases clear nucleation events are visible with particles growing up to approximately 150 nm in diameter





within the first 10 hours. As mentioned in Rosati et al. (2021a) an additional growth mode is observed and from Fig. 2 it seems to be more pronounced at dry compared to humid conditions (Fig. 2a vs. b).

$\kappa$-values are only presented for times when the selected dry diameter was within the boundary chosen to represent the main growth mode (see description in Sect. 2.3.4). The colour code describes the magnitude of $\kappa$, where dark blue colours represent

low values and light blue colours high values. Mean $\kappa$-values during the main particle growth mode for 80 nm particles (dry size) as retrieved from long-HTDMA measurements in Exp. 5 and 7 were $\kappa$=0.52 $\pm$ 0.03 and $\kappa$=0.54 $\pm$ 0.03, respectively. These were calculated from GF(80%) results of 1.42 $\pm$ 0.02 and 1.44 $\pm$ 0.02. $\kappa$-values derived for other particle sizes are presented in Table 2, and the GF(80%) results are presented in Table S3 (SI). In comparison $\kappa$-values from CCNc measurements for dry particles of approximately 80 nm yielded $\kappa$=0.50 $\pm$ 0.03 and $\kappa$=0.54 $\pm$ 0.06, for Exp. 5 and 7 respectively (all mean $\kappa$-

values from CCNc are presented in Table S4 in SI). Thus, for this size, mean $\kappa$-values at sub- and supersaturated water vapour conditions are well comparable and agree within the standard variations of the measurements. Figure S6 shows the derived mean $\kappa$-values as a function of all measured dry particle diameters for Exp. 5 and 7. There is no systematic trend in variation of $\kappa$ with particle diameter for $\kappa$-values calculated from GFs, while an increase in $\kappa$ with particle size can be seen for $\kappa$ values calculated from CCNc data. Such a size trend can be expected; for comparison the $\kappa$-value for ammonium sulphate varies from

0.51 to 0.63 over the particle size range 20-100 nm (calculated using UManSysprop at 293 K, Topping et al., 2016). Mean $\kappa$-values during the main particle growth mode are consistent across instruments and supersaturations. Agreement between $\kappa$-values derived for the same particle size from measurements at sub- and supersaturation is not necessarily expected as shown by several earlier studies for organic and inorganic particles (e.g. Petters and Kreidenweis, 2007; Whitehead et al., 2014; Pajunoja et al., 2015; Zhao et al., 2016; Rastak et al., 2017; Rosati et al., 2020).

Following the evolution of $\kappa$ for a certain size in Fig. 2 it can be seen that the values increase over time meaning that hygroscopicity increases with ageing by exposure to OH radicals and UV light indicating slight changes in chemical composition. In Exp. 7, performed at dry conditions (Fig. 2a), dry 80 nm particles could first be measured approximately 4 hours after the start of the experiment showing $\kappa$-values of 0.50, which increased to $\kappa$-values of 0.58 at about 7 hours after the start of the experiment. Similarly, in Exp. 5, performed at humid conditions (Fig. 2b), the $\kappa$-value for dry 80 nm particles changed from

0.51 to 0.63 (measured at approximately 4 and 8 hours after start of the experiment). Similar trends were observed in Exp. 8 and 10 performed at different temperatures and shown in SI.

To further investigate this change in hygroscopicity as a result of ageing, we extracted size-selected ePToF data from the HR-ToF-AMS. Figure 3 illustrates the temporal evolution of $m/z$ 79 and 80, representative for MSA and ammonium sulphate, respectively. Only data for particles with an aerodynamic diameter of approximately 126 nm are illustrated as these correspond

to particles with a mobility diameter of 80 nm assuming that particles were spherical and composed of approximately 50 % MSA and 50 % ammonium sulphate thus having a density of approximately 1.6 g cm$^{-3}$ (density of MSA: 1.48 g cm$^{-3}$, density of ammonium sulphate: 1.77 g cm$^{-3}$; according to Sigma-Aldrich). This assumption is based on bulk chemical composition data from HR-ToF-AMS presented in Fig. S7 (SI). Only results for Exp. 5 (wet conditions) and Exp. 6 (dry conditions) are shown, as the signal on the chosen marker ions was highest during these experiments and thus best PToF data analysis could

be performed. The black lines in Fig. 3 denote the ratio of $m/z$ 79 to 80 and the vertical grey, dashed lines depict the time





interval when this selected size was present within the main particle growth mode. When performing a linear regression to the ratio of *m/z* 79 to 80 when the ion signal was above 0.2 (threshold chosen to be less influenced by noise) a slow decrease is visible, indicating an increased fraction of ammonium sulphate in the particles with time (red solid lines in Fig. 3) under both dry and humid conditions. During this campaign the HR-ToF-AMS was not set-up in a way to resolve size-selected properties

at low aerosol mass concentration as only 30 s were spent on ePToF data acquisition for each 1 min data point. Thus, these results have to be treated with care and further experiments targeting size-selected chemical composition of secondary particles formed from the oxidation of DMS are needed to obtain a more detailed understanding of changes in the chemical composition of DMS derived products.

Previous HTDMA measurements of pure MSA-particles at 293 K yielded GF(90 %)=1.57 for a dry particle size of 100

nm (Johnson et al., 2004), which translates to $\kappa$=0.36 and GF(80 %)=1.33 (recalculated using $\kappa$-Köhler theory). Fossum et al. (2018) predicted the supersaturation needed to activate MSA particles at 293.15 K with dry sizes between 20 and 200 nm with the AIOMFAC model, finding a $\kappa$-value of 0.55 for 80 nm particles (critical supersaturations presented in SI of Fossum et al. (2018) and recalculated using $\kappa$-Köhler theory). Lower $\kappa$-values of 0.40 for 80 nm particles were modelled for ammonium-MSA (Fossum et al., 2018). In comparison $\kappa$-values for 80 nm ammonium sulphate particles are 0.59 and

0.62 according to the UManSysProp thermodynamic model (Topping et al., 2016) for subsaturated (GF(80 nm,80 %)) and supersaturated (80 nm dry size) conditions, respectively. Assuming particles composed of 50% MSA and 50% ammonium sulphate, as indicated by the HTR-ToF-MS results in this study, mixed $\kappa$-values can be retrieved using the modelled data of the single substances yielding 0.57 and 0.59 for 80 nm particles using ammonium sulphate results at sub- and supersaturated

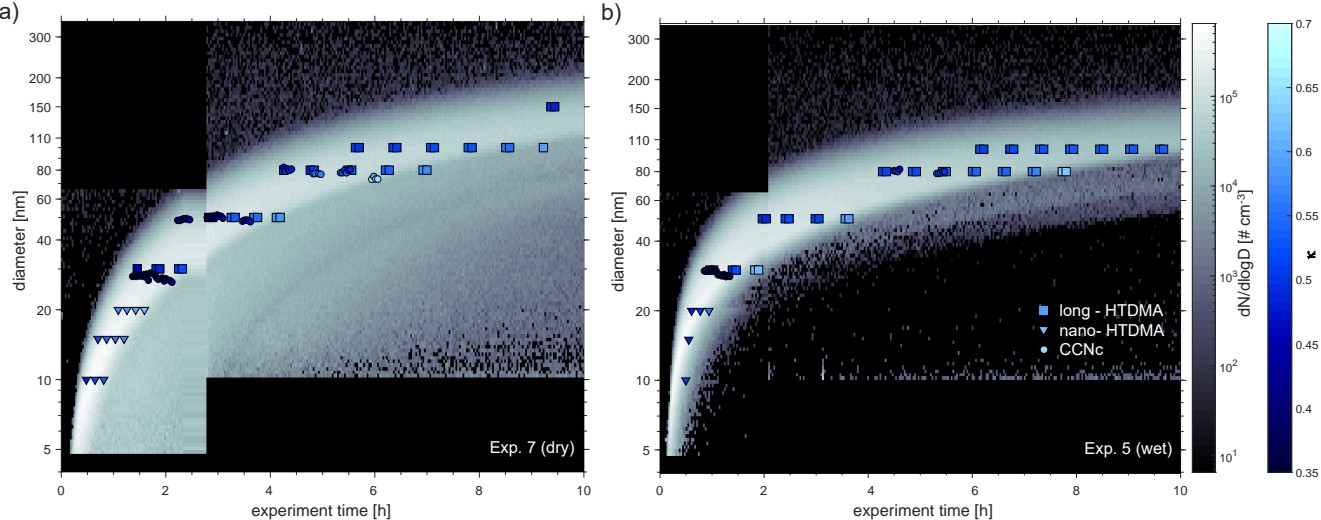

**Figure 2.** $\kappa$ values as obtained from nano- (triangles), long-HTDMA (squares) CCNc (circle) measurements for different dry particle sizes. Particle number size distributions of the DMS derived aerosol particles at dry and wet conditions are shown in the background. Results are from a) Exp. 7 (dry) and b) Exp. 5 (wet).





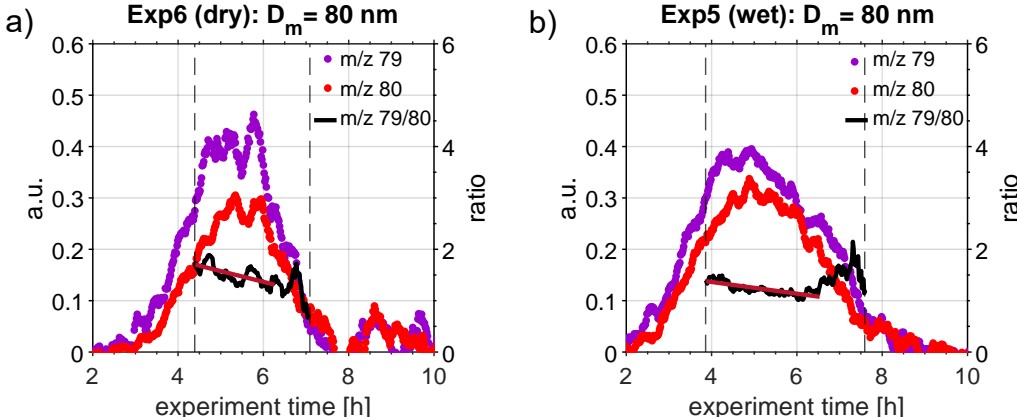

**Figure 3.** Temporal evolution of *m/z* 79 (violet dots) and 80 (red dots), representative for MSA and ammonium sulphate, respectively, as measured in the HR-ToF-AMS during a) Exp. 6 (dry conditions, T=293 K) and b) Exp. 5 (wet conditions, T=293 K). The black solid lines denote the ratio of *m/z* 79/80 as calculated during the main particle growth mode in the chamber (represented by vertical dashed lines). A linear fit was calculated for the ratio during the time period where both ions had a signal higher than 0.2 (dark red line), indicating a linear decrease over this period.

conditions, respectively. Our mean $\kappa$ results are considerably higher than those for pure MSA from HTDMA measurements presented earlier by Johnson et al. (2004) and closest to AIOMFAC $\kappa$-values for MSA while being lower than the ammonium sulphate values from UManSysProp. The range of $\kappa$-values measured within this study is consistent with the hygroscopicity of a mixture of MSA and ammonium sulphate. As the hygroscopicity of ammonium sulphate is larger than that of MSA, a change in chemical composition during ageing leading to a higher fraction of ammonium sulphate in the particles could be responsible for the measured increase in hygroscopicity over time.

### 3.2.2 Effect of Lower Temperatures

Experiments at 258, 273 and 293 K and high relative humidity were carried out to explore how the hygroscopicity and CCN activation potential is affected by the temperature during particle formation and subsequent ageing. Similar to the results at 293 K, the hygroscopicity of the particles formed at 258 and 273 K increases with ageing (see Fig. S8 and S9 in SI). Figure 4a illustrates mean GF(80 %) as measured by nano- and long-HTDMA and Fig. 4b mean $\kappa$-values as calculated from HTDMA and CCNc results for the three different temperatures. The mean values are calculated from the time period when a certain size was measured in the main growth mode as described in Sect. 2.3.4. Only data for humid experiments are shown, i.e. Exp. 2-5 and 8-11. Table 2 shows mean $\kappa$-values for each experiment (mean GF(80 %) values and mean $\kappa$-values from the CCNc are presented in Table S4 in the SI). When comparing results at 293 K in Fig. 4a, it is evident that GF(80 %) values are smaller for smaller sizes. This difference disappears when $\kappa$-values are compared (spherical symbols), indicating that this difference originates from the Kelvin term that is more important for particles of approximately 15 nm in dry size compared



**Table 2.** Mean $\kappa$-values and their standard deviations (sd) as calculated from nano- and long-HTDMA.

| Exp. # | $D_{dry}$ = 10 nm mean | sd | $D_{dry}$ = 15 nm mean | sd | $D_{dry}$ = 20 nm mean | sd | $D_{dry}$ = 30 nm mean | sd | $D_{dry}$ = 50 nm mean | sd | $D_{dry}$ = 80 nm mean | sd | $D_{dry}$ = 100 nm mean | sd | $D_{dry}$ = 150 nm mean | sd |
|---|---|---|---|---|---|---|---|---|---|---|---|---|---|---|---|---|
| 1 | 0.48 | 0.01 | 0.48 | 0.04 | 0.50 | 0.04 | 0.48 | 0.04 | 0.51 | 0.03 |  |  | 0.45 | 0.02 | 0.39 | 0.01 |
| 2 | 0.42 | 0.02 | 0.47 | 0.02 | 0.49 | 0.03 | 0.45 | 0.06 | 0.44 | 0.05 | 0.46 | 0.02 | 0.46 | 0.02 | 0.42 | 0.01 |
| 3 | 0.50 | 0.01 | 0.47 | 0.03 | 0.49 | 0.03 | 0.42 | 0.05 | 0.46 | 0.03 | 0.48 | 0.03 | 0.47 | 0.02 |  |  |
| 4 | 0.42 |  | 0.43 | 0.04 | 0.46 | 0.05 | 0.43 | 0.06 | 0.48 | 0.05 | 0.50 | 0.03 | 0.49 | 0.03 | 0.44 | 0.01 |
| 5 | 0.50 |  | 0.47 | 0.02 | 0.48 | 0.02 | 0.46 | 0.07 | 0.52 | 0.04 | 0.52 | 0.03 | 0.52 | 0.03 | 0.46 | 0.01 |
| 6 | 0.52 | 0.02 | 0.54 | 0.02 | 0.57 | 0.02 |  |  |  |  |  |  |  |  |  |  |
| 7 | 0.51 | 0.03 | 0.55 | 0.02 | 0.57 | 0.03 | 0.48 | 0.03 | 0.53 | 0.04 | 0.54 | 0.03 | 0.53 | 0.03 | 0.53 | 0.04 |
| 8 | 0.54 | 0.03 | 0.58 | 0.02 | 0.59 | 0.02 | 0.47 | 0.05 | 0.50 | 0.03 | 0.45 | 0.03 | 0.44 | 0.02 |  |  |
| 9 | 0.55 | 0.01 | 0.57 | 0.02 | 0.60 | 0.02 | 0.52 | 0.05 | 0.55 | 0.01 | 0.55 | 0.03 | 0.53 | 0.02 |  |  |
| 10 | 0.60 | 0.03 | 0.60 | 0.02 | 0.62 | 0.02 | 0.51 | 0.06 | 0.50 | 0.04 |  |  |  |  |  |  |
| 11 | 0.67 | 0.02 | 0.66 | 0.01 | 0.66 | 0.01 | 0.52 | 0.03 |  |  |  |  |  |  |  |  |

to approximately 80 nm. GF(80%) values in the long-HTDMA range seem comparable at all temperatures, while the nano-HTDMA shows a tendency for smaller GF(80%) at 293 K compared to 258 K. When comparing HTDMA data in Fig. 4b illustrating $\kappa$-values, this general trend of less hygroscopic particles at 293 K compared to 258 K is also visible. Figure 4b additionally illustrates $\kappa$-values as derived from CCNc measurements (square symbols). Generally, the ranges of the CCNc

results are more scattered but they overlap with HTDMA results. The majority of the herein presented experimental $\kappa$-values lie in the range of modelled values for MSA and ammonium-MSA as found by Fossum et al. (2018) (theoretical values depicted in Fig. 4b) and the experimental $\kappa$-values at 258 K for small particles exceed the modelled MSA results and are more comparable with the theoretical $\kappa$-values for ammonium sulphate.

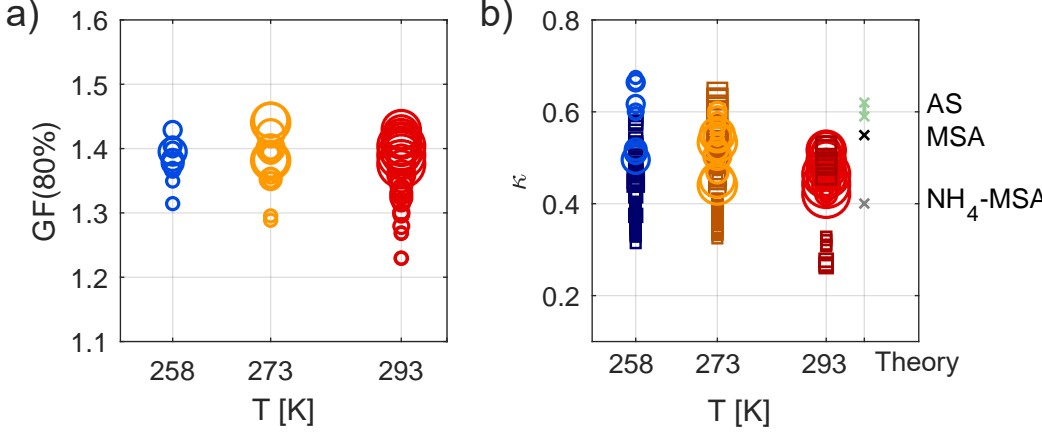

**Figure 4.** Temperature dependence of water uptake: a) GF(80 %) as measured from nano- and long-HTDMA (spheres), b) $\kappa$-values as measured from nano, long-HTDMA (spheres) and CCNc (squares). The size of the markers is related to the dry particle size selected, i.e. smaller markers describe results for smaller particles and vice versa. Results describe Exp. 2 - 5 and 8 - 11 at high humidity conditions. The theoretical $\kappa$ values are taken from Fossum et al. (2018) for MSA and NH$_4$MSA for 80 nm dry particles and from the UManSysProp model for ammonium sulphate (AS) as found for 80 nm dry particles as recalculated from sub- and supersaturated water vapour conditions.





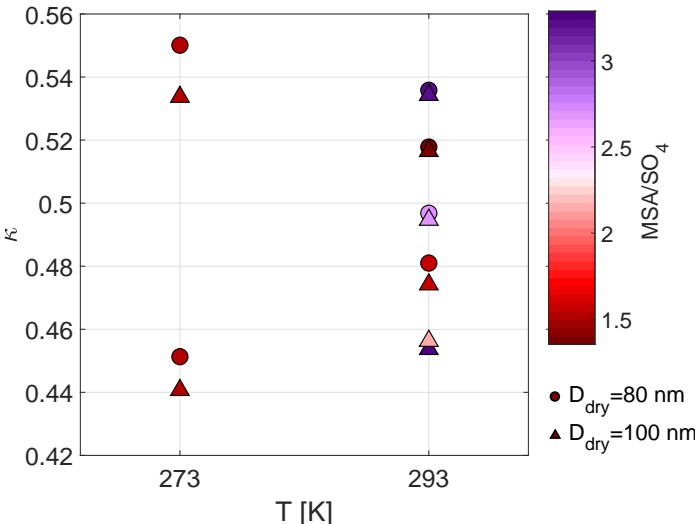

**Figure 5.** Mean $\kappa$-values at 273 K and 293 K for particles with dry sizes of 80 and 100 nm. Colour scheme shows the average MSA to sulphate ratio during the time period when the specific dry size was measured. Exp. 1-9 are shown.

The relative contribution of organics, nitrate, sulphate, ammonia, chloride and MSA are illustrated in Fig. S7 (SI) for ex-
periments performed at 273 and 293 K. The chemical composition of the aerosol particles could not be measured at 258 K
due to the low mass produced during these experiments. Exp. 8 and 9 were carried out at 273 K but no obvious difference
is found compared to the experiments carried out at 293 K. Figure 5 illustrates mean $\kappa$-values colour coded by the MSA to
sulphate ratio measured by the HR-ToF-AMS at 293 and 273 K. We selected $\kappa$-values for dry particles of 80 and 100 nm
(long-HTDMA) as these sizes are more representative of the size range analysed by the HR-ToF-AMS (typically PM$_1$). Tabu-
lar values of MSA/SO$_4^{2-}$ are given in Table S5 in the SI. The MSA/SO$_4^{2-}$ ratio is highest in Exp. 1 and 7, that were performed
at 293 K and dry conditions, reaching values above 3. These high values at dry and warm conditions were already discussed
in Wollesen de Jonge et al. (2021). Exp. 6, which was also carried out at 293 K and dry conditions, shows on the contrary a
small MSA/SO$_4^{2-}$ ratio of 1.3. This difference could potentially originate from the higher RH values of 13.7 $\pm$ 1.7 in Exp. 6
compared to RH below 10 % in Exp. 1 and 7. At 293 K and high humidity conditions the MSA/SO$_4^{2-}$ ratios vary from 2.7
to 1.4, where experiments at low H$_2$O$_2$ concentrations yield high values and experiments at high H$_2$O$_2$ conditions yield low
values. At 273 K and high humidity conditions the values during experiments with high or low H$_2$O$_2$ coincide yielding ratios
of 1.5. As seen in Fig. 5 the $\kappa$-values at 273 K vary most, while their MSA/SO$_4^{2-}$ ratios are the same. At 293 K, on the other
hand, the MSA/SO$_4^{2-}$ ratios vary from 3 to 1.4 as discussed above but no clear trend is observed in the $\kappa$-values, i.e. higher
$\kappa$-values are both associated with high and low MSA/SO$_4^{2-}$ ratios.

To the best of our knowledge there are no previous measurements of the water uptake of MSA or ammonium-MSA particles
at 258 or 273 K. When using the UManSysProp thermodynamic model to calculate $\kappa$-values for ammonium sulphate particles
at 293 and 258 K only small variations are found. $\kappa$-values of 0.59 and 0.58 at 293 and 258 K, respectively, are found based





on hygroscopic growth, while $\kappa$-values of 0.63 and 0.59 at 293 and 258 K, respectively, are found based on CCN activation

potential. This is consistent with previous findings showing that the water activity of ammonium sulphate in this temperature

range is quite constant (Gysel et al., 2002). Due to the similarities between ammonium sulphate and MSA we thus expect that

the temperature dependence of the water uptake of MSA is also negligible.

### 3.3 Quantum Chemical Calculations of Gas-phase Hydration of MSA

Gas-phase chemistry might be influenced by the presence of water molecules. We carried out detailed quantum chemical

calculations to investigate whether the formed MSA is hydrated in the gas-phase. Figure 6 presents the lowest free energy

cluster structures (298.15 K and 1 atm) obtained at the $\omega$B97X-D/6-31++G(d,p) level of theory.

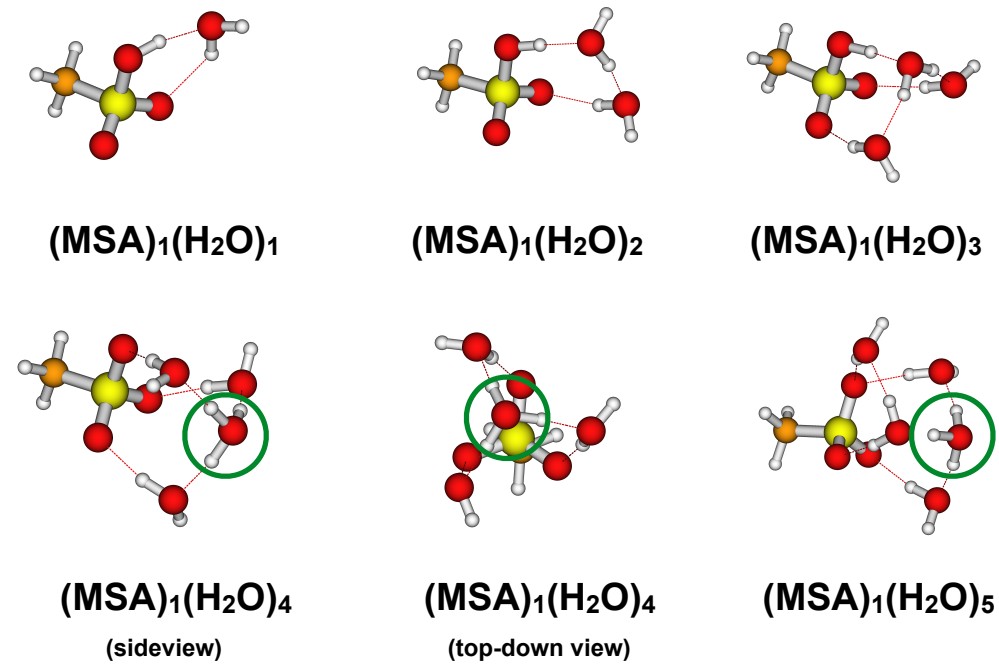

**Figure 6.** Identified lowest Gibbs free energy $(MSA)_1(H_2O)_{1-5}$ cluster structures calculated at the $\omega$B97X-D/6-31++G(d,p) level of theory. The green circles indicate the formation of a $H_3O^+$ ion.

The formed $(MSA)_1(H_2O)_{1-5}$ clusters show some differences compared to the corresponding sulphuric acid - water clusters

(see table S6). A proton transfer in the global free energy structures of the $(MSA)_1(H_2O)_{3-5}$ clusters is observed, whereas at

least six water molecules are required in the case of sulphuric acid clusters (Kildgaard et al., 2018a). The calculated reaction

free energy, at the DLPNO-CCSD(T$_0$)/aug-cc-pVTZ//$\omega$B97X-D/6-31++G(d,p) level of theory, for adding the first four water

molecules to MSA is less favourable than adding water molecules to sulfuric acid (see SI). This indicates that gas-phase MSA

is significantly less hygroscopic compared to SA.





The hydrate distribution ($x_n$) of the $(MSA)_1(H_2O)_{1-5}$ clusters can be calculated to elucidate the extent to which the clusters are hydrated under realistic atmospheric conditions. The hydrate distributions can be calculated as follows (Henschel et al., 2014a):

$$x_n = \left( \frac{p(H_2O)}{p_{ref}} \right)^n x_0 \exp \left( -\frac{\Delta G_n}{RT} \right). \tag{6}$$

Here $n$ is the given hydrate, $\Delta G_n$ is the binding free energy of the cluster and $p_{ref}$ is the reference pressure (1 atm). The population of the unhydrated cluster $x_0$ is normalised so that all the populations sum up to 1. $p(H_2O) = 0.0316$ atm is the saturation water vapour pressure at 298.15 K. We find that the MSA clusters are predominantly not hydrated in the gas-phase (see Figure S10 in SI). At low RH (20%) the MSA monohydrate population is only 3%. At increasing RH the monohydrate becomes more pronounced, with a value of 14% at 100% RH. Even at 100% RH the dihydrate is only 0.7% populated. In contrast, the monohydrate and dihydrate of SA is populated by 32% and 6%, respectively, at 100% RH (Kildgaard et al., 2018a). These findings imply that MSA is not significantly hydrated in the gas-phase and water should have little influence on the gas-phase chemistry. This is consistent with a previous study on the effect of a single water molecule on the reactions between the OH radical and MSA (Jörgensen et al., 2013), where water was also found to have no influence on the reaction kinetics.

## 4 Conclusions

The oceans emit large quantities of DMS, which can act as a precursor for the production and growth of aerosol particles. In this study we investigate the oxidation of DMS by hydroxyl radicals with a special focus on the hygroscopicity and CCN activity potential of the particles. The experiments were carried out in the temperature range 258 to 293 K at dry and humid conditions in the AURA chamber and complemented by quantum chemical calculations.

The nucleation and growth rates of the new particles formed were smaller at cold temperatures compared to at warm temperatures. This temperature dependence is unexpected but most likely originates from a slower oxidation of DMS at 258 K compared to at 293 K. Higher relative humidities did not seem to significantly affect the decay of DMS during oxidation with OH radicals but led to lower particle growth rates compared to at low relative humidities. Quantum chemical calculations revealed that gas-phase MSA clusters are predominantly not hydrated and thus high humidity conditions are not expected to influence its gas-phase chemistry.

The average hygroscopicity parameters, $\kappa$, calculated from measurements performed during the main particles' growth mode at sub- and supersaturated water vapour conditions at 293 K, ranged from 0.48 to 0.52 for dry 80 nm particles. These values are well comparable with modelled values for MSA or mixtures of MSA and ammonium sulphate (Fossum et al., 2018). These results are also consistent with online chemical composition measurements revealing that the largest fraction of the particles was MSA followed by ammonium sulphate. During ageing by exposure to hydroxyl radicals the water uptake of the particles increased, as found for all investigated sizes ($D_{dry}$=10 - 150 nm) at sub- and supersaturated water vapour conditions. The size selected results from the HR-ToF-AMS indicate that this could originate from a change in the MSA/sulphate ratio with ageing





favouring more ammonium sulphate in the particles with longer ageing. When oxidising DMS at colder temperatures, a slight
increase in $\kappa$-values is visible, which is more pronounced for smaller particles. The MSA/sulphate ratio did not show a clear
temperature trend but showed a dependency on RH, where highest values were measured at low relative humidities.

The $\kappa$-values found in this study for particles produced by the oxidation of DMS by OH radicals lie within the upper range of
the so-called "more-hygroscopic particle" (MHP) mode defined as $\kappa$ range between approximately 0.17 and 0.66 by Swietlicki
et al. (2008). This is the prevailing hygroscopicity group found from observations in the marine environment. Swietlicki et al.
(2008) described a potential source of these particles to be partly neutralised DMS-derived sulphate particles as well as aged sea
salt particles and mixtures with organic matter. Our hygroscopicity results are consistent with these observations underlining
the high water uptake potential of particles formed via the oxidation of DMS by OH radicals. Aerosol particles from new
particle formation have been shown to contribute the major fraction of all particles in remote continental regions (Spracklen
et al., 2006). Specifically at Arctic sites new particles formed from the oxidation of DMS have been found to be important
for the particle population, for their growth and for their potential to act as CCN (Beck et al., 2021; Schmale and Baccarini,
2021). Recently, Zheng et al. (2021) provided evidence for regular and frequent new particle formation in remote marine
locations thereby contradicting the common view of rare occurrences in these areas (e.g. Kerminen et al., 2018). There is still a
substantial knowledge gap of the detailed processes occurring in the ambient atmosphere. Thus, this laboratory study provides
new insights into the properties of particles formed from DMS oxidation at different temperatures and humidity regimes, which
is needed to understand atmospheric observations.

*Code and data availability.* The experimental data obtained in this study and the codes for the analysis, written in MATLAB, are available
on request from the first author B.R. Results from the quantum-chemical calculations are available on request from J.E.

*Author contributions.* B.R., M.B. and A.V designed and supervised the research; B.R., S.I. and S.C. performed and analysed the different
water uptake measurements; M.M.J and S.P.M. performed and analysed the chemical composition measurements with input from M.G.; B.R.
performed and analysed the gas-phase measurements; J.E. performed and analysed the quantum chemical simulations; A.M., A.V. and M.B.
contributed to interpreting the results. B.R. and S.I. wrote the paper with contributions from all co-authors. All authors read and reviewed the
manuscript.

*Competing interests.* The authors declare no conflict of interest.

*Acknowledgements.* This research was supported by the Austrian Science Fund (FWF: J 3970-N36), Aarhus University and Academy of
Finland Flagship funding (grant No 337550). The work of S.I. was financially supported by the University of Eastern Finland Doctoral
Program in Environmental Physics, Health and Biology. S.C. would like to thank the Carlsberg Foundation for this postdoc fellowship





(CF20-0637). We thank the Villum Foundation for support for the long-HTDMA involved in this study. We also thank Jon Bjarke Valbaek Mygind and Anders Feilberg for their involvement in the PTR measurements and Olli Väisänen for his help with the nano-HTDMA setup and analysis. We also thank Athanasios Nenes and his group for providing the SMCA analysis software.



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
