# Peer review of "Hygroscopicity and CCN potential of DMS derived aerosol particles"

_Atmospheric Chemistry and Physics, 2022_

## Author Comment (AC1)

**Review on "Hygroscopicity and CCN potential of DMS derived aerosol particles"**

We would like to thank both reviewers for their thorough and constructive comments. We appreciate the input and feel that the comments from the reviewers have helped improve clarity of our manuscript. We have addressed the reviewers' comments in a point-by-point fashion below and revised the manuscript accordingly. Our answers to the comments are given below in blue letters, while the referee comments are given in *black italics*. Additionally, we added the changes we made in the revised manuscript in **blue bold** letters.

**Reply to Reviewer: 1**

*The study by Rosati et al. investigated nucleation, hygroscopicity, and CCN activity of secondary aerosol particles from DMS as a function of temperature and relative humidity. Important findings include reduction in nucleation and particle growth rates at lower temperature (258 K) than that at 293 K. The kappa values for CCN activity were higher for lower temperature. Considering that temperature dependent particle formation from DMS has rarely been conducted, this study will serve as a good starting point for the more detailed studies in the future. The manuscript is well written, although some figures/tables can still be improved. I suggest publication of this manuscript after addressing the following comments.*

*Major comments:*

*1) Precursors of ammonium and nitrate:*
*The experimental procedure (section 2.2) describes that the authors injected H2O2 and DMS. Figure S7 suggests that particles generated in the chamber contained organics, nitrate, and ammonium, in addition to sulfate and MSA. It was not clear to me how ammonia and NOx concentrations were controlled. Ammonia is especially important both for nucleation and hygroscopic processes. So, the concentration of ammonia should clearly be summarized in Table 1. It seems that the authors measured NOx concentration using a NOx analyzer. It would be helpful for readers to understand the paper if the data from the instrument were to be available.*

Both $NO_x$ and ammonia are unfortunately always present in air, even after extensive cleaning. Thus, we could not completely avoid them. The $NO_x$ concentration was indeed monitored and the measured concentration range during an experiment is now added to the revised manuscript in Table 1. After acceptance of the manuscript all data and figures produced during the campaigns will be available to all researchers on sciencedata.dk.

We did not have an instrument to monitor the ammonia levels. However, measurements in Tange (56.35°N, 9.58°W; ~50 km from Aarhus; download from http://ebas.nilu.no/) show that ambient $NH_3$ concentrations were in the range of 1-7 ppb (up to approximately 5 µg/m$^3$ $NH_3$ as seen in the figure below) at this location during the experiment days. Thus, we expect that we had similar values inside the chamber during the experiments.

[Figure]

*Figure 1: Ammonia concentration at Tange during the time of experiments, downloaded from http://ebas.nilu.no*

We have added the following sentence to the revised manuscript:
**Ammonia concentrations during the experiments are estimated to be between 1-10 ppb based on data from Tange (56.352222°N, 9.5875°W; ~50 km from Aarhus; http://ebas.nilu.no/).**

2) *Formation mechanisms of organic compounds and its influence on hygroscopicity/CCN activity:*

*Figure S7 shows that the particles generated by the chamber contained 10 ~30 % of organics. I am wondering how they formed. It will be great if the authors could describe the corresponding formation mechanism. I also wondered if the organics could assist hygroscopic growth of aerosol particles. It would be great to have some additional discussion on this point.*

Organics were measured during each experiment and contributed to different % of the overall composition. Absolute concentrations were low in all experiments and organics had similar contributions during all experiments as illustrated in the figure below (Fig. 2).  We had seen an approximately constant contribution of organic mass in aerosols derived from the oxidation of DMS by hydroxyl radicals also in our previous work and discussed potential explanations therein (Rosati et al., 2021; Wollesen de Jonge et al., 2021).

In summary, it cannot be entirely excluded that despite careful cleaning the organic signal is due to a contamination in the chamber (e.g. from a previous experiment), it is however also known that the organic fraction as obtained from AMS spectra is susceptible to biases due to the complex fragmentation during ionization, uncertainties arising from the calibration with MSA, previously documented memory effects (e.g. Drewnick et al., 2015; Drewnick et al., 2009; Pieber et al., 2016) and the detection limit of the instrument.  The organic signal may thus also be an artefact related to the AMS and the associated data interpretation protocol. Since we do not know the type of organic it is difficult to access its potential impact on hygroscopicity.

[Figure]

*Figure 2: Absolute mass concentrations as measured from the AMS.*

The figure above is added to the supplementary of the revised manuscript.

Additionally, we have added the following text to the revised manuscript:
Line 374: **The relative contribution and absolute concentrations of organics, nitrate, sulphate, ammonia, chloride and MSA are illustrated in Fig. S7 (SI) for experiments performed at 273 and 293 K.**

Line 377: **Absolute concentrations show that in general the aerosol mass loading was small and that the organic contribution was comparable during all experiments. As previously discussed in Rosati et al. (2021) and Wollesen de Jonge et al. (2021) we cannot conclude on the exact origin of the organic signal. It cannot be entirely excluded that it is due to contamination from previous experiments. Since we do not know the type of organic it is difficult to access its potential impact on hygroscopicity. It could however also derive from biases due to the complex fragmentation during ionization, uncertainties arising from the calibration with MSA and previously documented memory effects (e.g. Drewnick et al., 2015; Drewnick et al., 2009; Pieber et al., 2016). If this is the case, the organic signal will not influence hygroscopicity.**

*Minor comments*
*Table 1: It is better to show H2O2 concentration by using mixing ratio.*

Assuming that the volume of the bag at the start of the experiment was 5 $m^3$ and that the density of the $H_2O_2$ solution (30% in $H_2O$) is 1.11 g $ml^{-1}$ the mixing ratios are:

| **418 µl** | 20 °C | 65.6 ppm | **1500 µl** | 20 °C | 235.5 ppm |
|---|---|---|---|---|---|
| | 0 °C | 61.2 ppm | | 0 °C | 219.5 ppm |
| | -15 °C | 57.8 ppm | | -15 °C | 207.4 ppm |

*Section 2.2.3: It seems that all the aerosol instruments were operated at room temperature, even though the chamber temperature was cooled down to 258 K. I wonder if potential changes in gas-particle equilibrium could be induced by the temperature change. I also wondered if the change in the equilibrium could influence the nucleation/particle growth rates in addition to chemical composition. It would be ideal to have some discussion about this point in the manuscript.*

Indeed, all instrumentation was operated outside the cooling chamber and thus at the standard laboratory temperature of 22°C.  Sampling lines for the AMS and SMPS were temperature insolated all the way to the instrument. We cannot fully exclude that some particle properties were affected by the difference in temperature between the bag and the laboratory.

We believe that the reported particle growth rates should not have been affected strongly as their residence time in the tubing before entering the nano or long-SMPS was between 11 – 28 s and growth rates, as illustrated in Fig. 1b, are in the order of nm h$^{-1}$.  The residence time in tubing to the PSM, used to measure the nucleation rates, was approximately 10 s. The fastest nucleation rates were measured during warm conditions and thus conditions where the laboratory and the chamber were comparable (see Fig. 1a). As the measured nucleation rates were slower at the colder temperatures, we expect changes to be slower and thus less affected by the time the particles spent in the warmer laboratory.

We have added the following sentence to the manuscript:
Line 122: **All instruments were operated at the laboratory temperature of 22°C outside the AURA chamber. To limit evaporation in tubing en route to the particle size (SMPS) and chemical composition measurement (AMS), the tubing was insulated.**

*Section 2.3.5.: The authors mention that the HTDMA measurement was conducted at RH = 80%. Deliquescence relative humidity of ammonium sulfate is slightly lower than 80%. Considering the chemical composition of the particles, I personally think that it would have been better to conduct the HTDMA measurement at higher RH. Could the authors explain the reason why they selected this condition?*

RH=80% was chosen because it could repeatably and stably be set at the laboratory conditions. Particularly in the long-HTDMA higher RH led to quite large fluctuations and thus less stable conditions. We calibrated the HTDMAs with pure ammonium sulfate (see result for long-HTDMA in figure below) and could clearly see that the deliquescence was around 78% and all particles were liquid at RH=80%. We also looked closely at the shape of the humid size distributions to see whether we had some bimodal distributions as often found when too close to the deliquescence (e.g. Biskos et al., 2006) but this was never the case.

[Figure]

*Figure 3: Humidogram for Ammonium Sulfate particles with a dry size of 200 nm.*

*Figure 1a: There is a cyclic oscillation in the nucleation rate at 273 K. Is it possible to explain the potential cause?*

Yes, as stated in the caption of Fig. 1, the fluctuations stem from the fact that the PSM was set to measure in scanning mode after a while. This leads to small changes in the sample flow rate that then induce these fluctuations.

*L309 Such a size trend can be expected: I could not understand the reason why it can be expected. Please describe it in more detail.*

It is expected from Köhler theory. κ has a size dependence because the water activity at activation depends on the initial dry particle size.

For example: the κ-value for ammonium sulphate varies from 0.51 to 0.63 over the particle size range 20-100 nm (calculated using UManSysprop at 293 K, Topping et al., 2016).

We have reformulated the text to make this clearer:
Line 309 and following: **Such a size trend can be expected from Köhler theory; for comparison the κ-value for ammonium sulphate varies from 0.51 to 0.63 over the particle size range 20-100 nm.**

*Figure 2: It requires a lot of efforts and focus to see this figure. I suggest updating the figure so that the readers can easily get the main message of the figure.*

We have spent a substantial amount of time discussing other versions of this figure but to our mind this is still the best way to illustrate the data. We have though changed the color-code and increased the size of the text and markers in order to make it easier to read.

[Figure]

*Table 2: The formats of tables 1 and 2 are significantly different. It would be better to have a standardized table format for a manuscript.*

The tables were re-formatted to have the same table format in the revised manuscript.

**Reply to Reviewer: 2**

*This work mainly investigated the hygroscopicity and cloud condensation nuclei activity of aerosol particles formed through oxidation of DMS. DMS is emitted by phytoplankton species in the oceans and constitutes the largest fraction of naturally emitted sulphur to the atmosphere. Secondary aerosols formed through oxidation of DMS play significant roles in climate and their hygroscopic properties are key parameters in describing their direct and indirect climate effects. Thus, scientific findings of this study are meaningful and fits well the scope of ACP. The authors have designed comprehensive laboratory chamber experiments and performed very good measurements using a suite of aerosol instruments, especially including synthesized nano-HTDMA, HTDMA, CCN and AMS measurements. In general, experiments are well designed and discussions are insightful. I have one major concern and some minor suggestions as listed below:*

*Major comment:*

*As shown in Fig.S7, organic constitutes a non-negligible fraction to formed aerosol mass, but compositions and impacts of organics on hygroscopicity are not even mentioned in the discussions. It seems that authors just neglected this part of aerosol mass, and compared measured aerosol hygroscopicity only with mixed MSA-sulfate system. Authors should explain a little bit their choice in discussions or add more discussions. Also, it would be nice to discuss possible compositions of formed organics.*

Fig. S7 shows that the relative contribution of organics can sometimes be as high as 30% but in absolute terms the contribution is actually quite constant throughout the measurement period (see figure below).

[Figure]

The figure shown above is added to the supplementary material of the revised manuscript.

Organics were measured during each experiment and contributed to different % of the overall composition. Absolute concentrations were low in all experiments and organics had similar contributions during all experiments as illustrated in the figure below (Fig. 2). We had seen an approximately constant contribution of organic mass in aerosols derived from the oxidation of DMS by hydroxyl radicals also in our previous work and discussed potential explanations therein (Rosati et al., 2021; Wollesen de Jonge et al., 2021).

In summary, it cannot be entirely excluded that despite careful cleaning the organic signal is due to a contamination in the chamber (e.g. from a previous experiment), it is however also known that the

organic fraction as obtained from AMS spectra is susceptible to biases due to the complex fragmentation during ionization, uncertainties arising from the calibration with MSA, previously documented memory effects (e.g. Drewnick et al., 2015; Drewnick et al., 2009; Pieber et al., 2016) and the detection limit of the instrument.  The organic signal may thus also be an artefact related to the AMS and the associated data interpretation protocol. Since we do not know the type of organic it is difficult to access its potential impact on hygroscopicity.

Additionally, we have added the following text to the revised manuscript:
Line 374: **The relative contribution and absolute concentrations of organics, nitrate, sulphate, ammonia, chloride and MSA are illustrated in Fig. S7 (SI) for experiments performed at 273 and 293 K.**

Line 377: **Absolute concentrations show that in general the aerosol mass loading was small and that the organic contribution was comparable during all experiments. As previously discussed in Rosati et al. (2021) and Wollesen de Jonge et al. (2021) we cannot conclude on the exact origin of the organic signal. It cannot be entirely excluded that it is due to contamination from previous experiments. Since we do not know the type of organic it is difficult to access its potential impact on hygroscopicity. It could however also derive from biases due to the complex fragmentation during ionization, uncertainties arising from the calibration with MSA and previously documented memory effects (e.g. Drewnick et al., 2015; Drewnick et al., 2009; Pieber et al., 2016).  If this is the case, the organic signal will not influence hygroscopicity.**

*Minor suggestions:*
*L105: More details about Kappa calculations from CCN measurements should be given*

We have added the following sentence to the revised manuscript:
L 101 and following:

**To compare water uptake at sub- and supersaturated conditions, the hygroscopicity parameter κ, introduced by Petters and Kreidenweis (2007), was used. The water activity term in Eq. 2 is be described in terms of κ in the following way:**
$$\frac{1}{a_w} = 1 + \kappa \frac{V_s}{V_w} \qquad (4)$$
**where $V_s$ is the volume of the dry particle and $V_w$ the volume of the water in the droplet. By combining Eq. 3, 4 and 5 the semi-empirical κ-Köhler theory can be obtained.**

*L121: "further"-> "already"?*

Yes, that is correct. We changed the wording in the revised version.

*Section 2.3.5: set-up of the two HTDMAs are overall similar, introductions of them summarized in one paragraph might be better*

We combined the paragraphs with the HTDMA description.

*Section 2.4: I suggest present a brief introduction of quantum chemical calculations and needed input parameters before 2.4.1.*

We agree that a brief introduction to the quantum chemical calculations should be added. As these are first principal calculations, we do not need other input parameters except the xyz coordinates of

MSA and a water. The cluster structures of the MSA-water clusters are then obtained via geometry optimization. All the necessary information on how the calculations were conducted are contained in section 2.41, 2.42 and the SI.

We have added the following text to introduce the quantum chemical calculations in the manuscript:
**To obtain molecular level insight into the hydration free energy of MSA, we carried out highly accurate quantum chemical calculations of the MSA-hydrate cluster structures. These calculations will aid in elucidating to what extent the MSA molecules are hydrated in the gas-phase.**

*L268: More discussions? Elucidate why it is interesting*

Unfortunately, we do not know the reason why different temperature dependences of the rate constant for the reaction of DMS with OH have been reported in the literature and an expanded investigation of the reason is outside the scope of the current work.

We have reformulated the text to read:

**The decrease of nucleation and growth rates with decreasing temperature is consistent with a lower observed loss rate of DMS at lower temperatures, see Fig. S4 and Table S1 in SI illustrating an example for each of the temperatures investigated in this study and a comparison between dry and humid experiments (Exp. 1, 2, 8 and 10). Different temperature trends for the rate of the reaction of DMS with OH radicals have been reported in the literature (Albu et al., 2006; Barnes et al., 2006; Hoffmann et al., 2016). Our observed trend is in line with the predicted temperature dependence obtained with the Arrhenius expression used in the Master Chemical Mechanism version 3.3.1 (MCMv3.3.1) and Hoffmann et al. (2016) delineating slower reaction rates at colder temperatures.**

*L282: "higher absolute concentration", higher-> lower?*

We thank the reviewer for spotting this mistake!

We have reformulated the text in the following way:

**The absolute water concentration could also have an effect on the observed trends in the nucleation rate as the initial cluster formation in our chamber setup most likely consist of relatively weakly bound sulphuric acid – methane sulfonic acid – ammonia clusters (Rosati et al., 2021). Water has been shown to increase the nucleation rate if the initial cluster is weakly bound such as for sulphuric acid – ammonia clusters (Henschel et al., 2016; Henschel et al., 2014). However, in our studies the absolute concentration of water is lowest at colder temperatures.**

**References:**

Albu, M., Barnes, I., Becker, K. H., Patroescu-Klotz, I., Mocanu, R., & Benter, T. (2006). Rate coefficients for the gas-phase reaction of OH radicals with dimethyl sulfide: temperature and O2 partial pressure dependence [10.1039/B512536G]. *Physical Chemistry Chemical Physics*, *8*(6), 728-736. https://doi.org/10.1039/B512536G

Barnes, I., Hjorth, J., & Mihalopoulos, N. (2006). Dimethyl Sulfide and Dimethyl Sulfoxide and Their Oxidation in the Atmosphere. *Chemical Reviews*, *106*(3), 940-975. https://doi.org/10.1021/cr020529+

Biskos, G., Malinowski, A., Russell, L. M., Buseck, P. R., & Martin, S. T. (2006). Nanosize Effect on the Deliquescence and the Efflorescence of Sodium Chloride Particles. *Aerosol Science and Technology*, *40*(2), 97-106. https://doi.org/10.1080/02786820500484396

Drewnick, F., Diesch, J. M., Faber, P., & Borrmann, S. (2015). Aerosol mass spectrometry: particle–vaporizer interactions and their consequences for the measurements. *Atmos. Meas. Tech.*, *8*(9), 3811-3830. https://doi.org/10.5194/amt-8-3811-2015

Drewnick, F., Hings, S. S., Alfarra, M. R., Prevot, A. S. H., & Borrmann, S. (2009). Aerosol quantification with the Aerodyne Aerosol Mass Spectrometer: detection limits and ionizer background effects. *Atmos. Meas. Tech.*, *2*(1), 33-46. https://doi.org/10.5194/amt-2-33-2009

Henschel, H., Kurtén, T., & Vehkamäki, H. (2016). Computational Study on the Effect of Hydration on New Particle Formation in the Sulfuric Acid/Ammonia and Sulfuric Acid/Dimethylamine Systems. *The Journal of Physical Chemistry A*, *120*(11), 1886-1896. https://doi.org/10.1021/acs.jpca.5b11366

Henschel, H., Navarro, J. C. A., Yli-Juuti, T., Kupiainen-Määttä, O., Olenius, T., Ortega, I. K., Clegg, S. L., Kurtén, T., Riipinen, I., & Vehkamäki, H. (2014). Hydration of Atmospherically Relevant Molecular Clusters: Computational Chemistry and Classical Thermodynamics. *The Journal of Physical Chemistry A*, *118*(14), 2599-2611. https://doi.org/10.1021/jp500712y

Hoffmann, E. H., Tilgner, A., Schrödner, R., Bräuer, P., Wolke, R., & Herrmann, H. (2016). An advanced modeling study on the impacts and atmospheric implications of multiphase dimethyl sulfide chemistry. *Proceedings of the National Academy of Sciences*, *113*(42), 11776-11781. https://doi.org/doi:10.1073/pnas.1606320113

Pieber, S. M., El Haddad, I., Slowik, J. G., Canagaratna, M. R., Jayne, J. T., Platt, S. M., Bozzetti, C., Daellenbach, K. R., Fröhlich, R., Vlachou, A., Klein, F., Dommen, J., Miljevic, B., Jiménez, J. L., Worsnop, D. R., Baltensperger, U., & Prévôt, A. S. H. (2016). Inorganic Salt Interference on CO2+ in Aerodyne AMS and ACSM Organic Aerosol Composition Studies. *Environmental Science & Technology*, *50*(19), 10494-10503. https://doi.org/10.1021/acs.est.6b01035

Rosati, B., Christiansen, S., Wollesen de Jonge, R., Roldin, P., Jensen, M. M., Wang, K., Moosakutty, S. P., Thomsen, D., Salomonsen, C., Hyttinen, N., Elm, J., Feilberg, A., Glasius, M., & Bilde, M. (2021). New Particle Formation and Growth from Dimethyl Sulfide Oxidation by Hydroxyl Radicals. *ACS Earth and Space Chemistry*, *5*(4), 801-811. https://doi.org/10.1021/acsearthspacechem.0c00333

Topping, D., Barley, M., Bane, M. K., Higham, N., Aumont, B., Dingle, N., & McFiggans, G. (2016). UManSysProp v1.0: an online and open-source facility for molecular property prediction and atmospheric aerosol calculations. *Geosci. Model Dev.*, *9*(2), 899-914. https://doi.org/10.5194/gmd-9-899-2016

Wollesen de Jonge, R., Elm, J., Rosati, B., Christiansen, S., Hyttinen, N., Lüdemann, D., Bilde, M., & Roldin, P. (2021). Secondary aerosol formation from dimethyl sulfide – improved mechanistic understanding based on smog chamber experiments and modelling. *Atmos. Chem. Phys.*, *21*(13), 9955-9976. https://doi.org/10.5194/acp-21-9955-2021

---

## Author Response (AR2)

**Review on "Hygroscopicity and CCN potential of DMS derived aerosol particles"**

We would like to thank the editor for his constructive comments. We have addressed the comments in a point-by-point fashion below and revised the manuscript accordingly. Our answers to the comments are given below in blue letters, while the editor comments are given in *black italics*. Additionally, we added the changes we made in the revised manuscript in **blue bold** letters.

**(1)** *Both reviewers pointed out the issue of having 0.5-1 ug m-3 non-MSA OA detected by the AMS and its unclear influence on hygroscopicity of the generated particles. The mass fractions of such OA in some experiments are high. If the OA was from contamination, a high mass fraction of it may affect the measured hygroscopicity, e.g., when the OA is insoluble. If it was from analysis bias, should this part of mass be included into the MSA mass? I think what was added in Line 377 remains unclear and is insufficient to address reviewers' concerns. The key question is whether this mass should be added or excluded in the analysis herein. I would like to see more discussion with additional evidence on it. For example, the mass spectra of OA should provide some hints on the source and properties of OA.*

We agree with the editor that the OA signal is in some cases quite high and that the determination of the origin would be very helpful to understand its importance in the experiments. In order to discuss this further we present here some additional data for Exp. 5 to give an example.

Fig. R1 and Fig. R2 illustrate the organic mass spectra before DMS was injected into the chamber and when the particle mass reached its highest concentration, respectively. From Fig. R1 it is evident that the chamber is clean with respect to organic particle mass before DMS is introduced. The $CH_4O^+$ and the negative $CH_2^+$ signals are both artifacts of peak fitting in normal AMS data processing. m/z indicated in orange in Fig. R1 and R2 denote peaks that were also found during the MSA calibration (MSA calibration spectrum is shown in Fig. R3). In Fig. R2 some other ions appear to have negative concentrations pointing to the fact that too much of the signal of those specific ions was attributed to MSA rather than organics. This shows that the MSA fragmentation or ion transfer during the calibration was slightly different from MSA fragmentation and ion transfer under chamber conditions. This is within normal uncertainty for AMS. Looking closer at the other ions in the spectrum we found that $CH_4SO_2^+$ (indicated in blue), could stem from MSIA. There is also a potential contribution from dimethyl sulfoxide (DMSO) on ion $C_2H_6SO^+$. The signal is though so low it not clearly visible in Fig. R2. $C_6H_5^+$ and $C_7H_7^+$ are typical markers for aromatic compounds but represent only a small contribution of the overall organic mass. Finally, the organic mass signal is composed of the sum of many low-intensity fragments that we cannot attribute to a specific source. In comparison Fig. R4 shows the residual organic mass signal recorded during the MSA calibration, after the verified MSA ions (shown in Fig. R3) are subtracted. The signal intensities are clearly lower compared to during DMS oxidation experiments in the chamber and fewer peaks are visible (Fig. R3 vs. R4). Nonetheless, this also illustrates that even during calibration some organic fragments appear that we cannot clearly assign to a specific source.

[Figure]

Fig. R1: High resolution mass spectrum of the organic signal measured during Exp. 5 before DMS was injected into the chamber. The ions indicated in orange color denote ions also found during the MSA calibration.

[Figure]

Fig. R2: High resolution mass spectrum of the organic signal measured during the time period where the particle mass peaked in Exp. 5. The ions indicated in orange color denote ions also found during the MSA calibration. Ions denoted in black color have an unknown origin. The ion denoted in blue could stem from MSIA.

[Figure]

Fig. R3: High resolution mass spectrum of MSA from calibration including only verified MSA ions (Hodshire et al., 2019).

[Figure]

Fig. R4: High resolution mass spectrum of residual organics, i.e., unaccounted organics after subtracting verified MSA signal, at about 1.5 µg/m³ MSA during calibration.

In order to assess the potential role of organic compounds for hygroscopicity we have performed a κ-sensitivity calculation:

According to Petters and Kreidenweis (2007) the simple volume weighted mixing rule can be applied to retrieve the mixed hygroscopicity parameter κ from organic and inorganic compounds. In order to

assess the importance of a potential organic contribution to the measured κ (κ$_{meas}$) we can use the following equation (mixing rule):

$$\kappa_{meas} = \kappa_{org} \cdot \varepsilon_{org} + \kappa_{mix,in} \cdot \varepsilon_{mix,in} \tag{1}$$

Where κ$_{org}$ is the κ of the organic compounds, κ$_{mix,in}$ is the effective κ of the mixture of all inorganic contributions and ε$_{org}$ and ε$_{mix,in}$ are the volume fractions of the organics and inorganics, respectively. The volume fractions of organics and inorganics can be calculated from the mass concentrations measured by the HR-ToF-MS and assuming densities of the organics and inorganics, respectively. For organic compounds density values of approximately 1.2-1.4 g cm$^{-3}$ have been suggested (Nakao et al., 2013). For the expected inorganic compounds (ammonium sulfate, MSA and ammonium-MSA) Fossum et al. (2018) suggest densities of approximately 1.7, 1.5 and 1.4 g cm$^{-3}$ for ammonium sulfate, MSA and ammonium-MSA, respectively. By assuming a combination of these three inorganic compounds we estimate an average inorganic density of approximately 1.6 g cm$^{-3}$. Additionally, we assume the κ of the organics (κ$_{org}$) to be between 0 and 0.2 based on Jimenez et al. (2009). By using these inputs and rearranging Eq.1, the actual κ of the inorganic compounds during the measurements (κ$_{mix,in}$) can be found as:

$$\kappa_{mix,in} = \frac{\kappa_{meas} - \kappa_{org} \cdot \varepsilon_{org}}{\varepsilon_{mix,in}} \tag{2}$$

For Exp.5, where mass spectra are presented above, κ$_{mix,in}$ in the range 0.6 – 0.7 are obtained when using κ$_{meas}$=0.52 (mean value for dry 80 nm particles) and κ$_{org}$ between 0 – 0.2. This would represent a κ of pure ammonium sulfate particles. For Exp.1, where organic contribution is highest, a κ$_{mix,in}$ in the range 0.7 – 0.8 is found when using κ$_{meas}$=0.45 (mean value for dry 80 nm particles). These values exceed the κ-range of pure ammonium sulfate. Both cases contradict the results from the HR-ToF-MS that did not only find an ammonium sulfate contribution but a considerable fraction of MSA in all experiments (see Fig. S7a and b; MSA fraction always above 37 %). This MSA fraction is expected to lower the overall inorganic κ, as its predicted κ value is 1.5 (Fossum et al., 2018).

These calculations thus point towards organics rather being a measurement artefact. Our data and the assumptions for the above presented hygroscopicity calculations do not make it possible to conclude on whether the organic mass should be added or excluded in the analysis. Nevertheless, we can conclude that the presented κ$_{meas}$ values represent lower limits of κ-values as any contribution of organic compounds (0< κ$_{org}$< 0.2) would imply a higher κ-value from the inorganic fraction.

We have revised the added text in Line 377 as follows:

**Absolute concentrations show that in general the aerosol mass loading was small and that the organic contribution was comparable during all experiments. As previously discussed in Rosati et al. (2021) and Wollesen de Jonge et al. (2021) we cannot conclude on the exact origin of the organic signal. A thorough analysis of the organic mass spectra revealed that the organic mass signal is composed of the sum of many low-intensity fragments that we cannot attribute to a specific source (see example organic mass spectrum for Exp. 5 in SI). It cannot be entirely excluded that it is due to contamination from previous experiments. It could however also derive from biases due to the complex fragmentation during ionization and uncertainties arising from the calibration with MSA. Additionally, instrumental memory effects have previously been documented for the AMS (e.g. Drewnick et al., 2015; Drewnick et al., 2009; Pieber et al., 2016). Two ionization efficiency calibrations before and after the experiments could neither exclude nor completely prove memory effects. If the organic signal stems from uncertainties in the analysis, it**

will not influence hygroscopicity. In the case that the organic signal is real, the herein presented hygroscopicity results represent lower estimates of the inorganic aerosol hygroscopicity as κ-values for organics in the range 0-0.2 (Jimenez et al., 2009) can be expected, which would lead to a decrease of the κ-values measured for an organic-inorganic mixture.

The abstract has been modified to read:

**"We show that the hygroscopicity parameter for particles of 80 nm in diameter is in the range 0.48 - 0.52 or higher, at 293 K as measured at both sub- and supersaturated water vapour conditions."**

The conclusion has been modified to read:

**The average hygroscopicity parameters, κ, calculated from measurements performed during the main particles' growth mode at sub- and supersaturated water vapour conditions at 293 K, ranged from 0.48 to 0.52 for dry 80 nm particles. These values may represent lower limits and are well comparable with modelled values for MSA or mixtures of MSA and ammonium sulphate (Fossum et al., 2018).**

**(2)** *The caption of Figure 1 stated that the fluctuations (I guess after 0.4 h should also be stated) are caused by scanning-mode operation of PSM. Are there any reference to support? Also, Sect. 2.3.3 is confusing because it said the instrument was operated in a fixed mode. The question is if the scanning mode is problematic, can we trust the data in Fig. 1 for that part? Please update Sect. 2.3.3 and describe PSM operation as well as its problem in details.*

The fluctuations caused by setting the PSM to scanning mode after about 0.4 h for the temperature of 273 K, do not have any effect on the calculations of the nucleation rates (described in Sect. 2.1.1). There is no problem or issue of the PSM in scanning mode, this is the normal way it behaves in this mode (Lehtipalo et al., 2014). This part of the measurement was shown purely to illustrate that the PSM had reached a maximum. The data when the PSM was measuring in scanning mode are not used in any calculations in this manuscript. To make this point clearer we have adapted the figure in the following way:

[Figure]

[Figure]

Additionally, we have added this sentence to the figure caption:

**The fluctuations seen in at T=273 K (light yellow dots) are due to the fact that the PSM was set to measure in scanning mode during this period (Lehtipalo et al., 2014). This data was not used for the**

calculation of $J_{1.7}$ **and is only illustrated to show the reader that the maximum in number concentration had been reached.**

**References:**

Drewnick, F., Diesch, J. M., Faber, P., & Borrmann, S. (2015). Aerosol mass spectrometry: particle–vaporizer interactions and their consequences for the measurements. *Atmos. Meas. Tech.*, *8*(9), 3811-3830. https://doi.org/10.5194/amt-8-3811-2015

Drewnick, F., Hings, S. S., Alfarra, M. R., Prevot, A. S. H., & Borrmann, S. (2009). Aerosol quantification with the Aerodyne Aerosol Mass Spectrometer: detection limits and ionizer background effects. *Atmos. Meas. Tech.*, *2*(1), 33-46. https://doi.org/10.5194/amt-2-33-2009

Fossum, K. N., Ovadnevaite, J., Ceburnis, D., Dall'Osto, M., Marullo, S., Bellacicco, M., Simó, R., Liu, D., Flynn, M., Zuend, A., & O'Dowd, C. (2018). Summertime Primary and Secondary Contributions to Southern Ocean Cloud Condensation Nuclei. *Scientific Reports*, *8*(1), 13844. https://doi.org/10.1038/s41598-018-32047-4

Hodshire, A. L., Campuzano-Jost, P., Kodros, J. K., Croft, B., Nault, B. A., Schroder, J. C., Jimenez, J. L., & Pierce, J. R. (2019). The potential role of methanesulfonic acid (MSA) in aerosol formation and growth and the associated radiative forcings. *Atmos. Chem. Phys.*, *19*(5), 3137-3160. https://doi.org/10.5194/acp-19-3137-2019

Jimenez, J. L., Canagaratna, M. R., Donahue, N. M., Prevot, A. S. H., Zhang, Q., Kroll, J. H., DeCarlo, P. F., Allan, J. D., Coe, H., Ng, N. L., Aiken, A. C., Docherty, K. S., Ulbrich, I. M., Grieshop, A. P., Robinson, A. L., Duplissy, J., Smith, J. D., Wilson, K. R., Lanz, V. A., . . . Worsnop, D. R. (2009). Evolution of Organic Aerosols in the Atmosphere. *Science*, *326*(5959), 1525-1529. https://doi.org/10.1126/science.1180353

Lehtipalo, K., Leppä, J., Kontkanen, J., Kangasluoma, J., Franchin, A., Wimnner, D., Schobesberger, S., Junninen, H., Petäjä, T., & Sipilä, M. (2014). Methods for determining particle size distribution and growth rates between 1 and 3 nm using the Particle Size Magnifier. *Boreal Environment Research*.

Nakao, S., Tang, P., Tang, X., Clark, C. H., Qi, L., Seo, E., Asa-Awuku, A., & Cocker, D. (2013). Density and elemental ratios of secondary organic aerosol: Application of a density prediction method. *Atmospheric Environment*, *68*, 273-277. https://doi.org/https://doi.org/10.1016/j.atmosenv.2012.11.006

Petters, M. D., & Kreidenweis, S. M. (2007). A single parameter representation of hygroscopic growth and cloud condensation nucleus activity. *Atmos. Chem. Phys.*, *7*(8), 1961-1971. https://doi.org/10.5194/acp-7-1961-2007

Pieber, S. M., El Haddad, I., Slowik, J. G., Canagaratna, M. R., Jayne, J. T., Platt, S. M., Bozzetti, C., Daellenbach, K. R., Fröhlich, R., Vlachou, A., Klein, F., Dommen, J., Miljevic, B., Jiménez, J. L., Worsnop, D. R., Baltensperger, U., & Prévôt, A. S. H. (2016). Inorganic Salt Interference on CO2+ in Aerodyne AMS and ACSM Organic Aerosol Composition Studies. *Environmental Science & Technology*, *50*(19), 10494-10503. https://doi.org/10.1021/acs.est.6b01035

Rosati, B., Christiansen, S., Wollesen de Jonge, R., Roldin, P., Jensen, M. M., Wang, K., Moosakutty, S. P., Thomsen, D., Salomonsen, C., Hyttinen, N., Elm, J., Feilberg, A., Glasius, M., & Bilde, M. (2021). New Particle Formation and Growth from Dimethyl Sulfide Oxidation by Hydroxyl Radicals. *ACS Earth and Space Chemistry*, *5*(4), 801-811. https://doi.org/10.1021/acsearthspacechem.0c00333

Wollesen de Jonge, R., Elm, J., Rosati, B., Christiansen, S., Hyttinen, N., Lüdemann, D., Bilde, M., & Roldin, P. (2021). Secondary aerosol formation from dimethyl sulfide – improved mechanistic understanding based on smog chamber experiments and modelling. *Atmos. Chem. Phys.*, *21*(13), 9955-9976. https://doi.org/10.5194/acp-21-9955-2021